# Oxidized sulfur-rich arc magmas formed porphyry Cu deposits by 1.88 Ga

Xuyang Meng [1✉], Jackie M. Kleinsasser[2], Jeremy P. Richards[1,8], Simon R. Tapster [3], Pedro J. Jugo [1], Adam C. Simon[2], Daniel J. Kontak[1], Laurence Robb [4,5], Grant M. Bybee[6], Jeffrey H. Marsh[1] & Richard A. Stern [7]

Most known porphyry Cu deposits formed in the Phanerozoic and are exclusively associated with moderately oxidized, sulfur-rich, hydrous arc-related magmas derived from partial melting of the asthenospheric mantle metasomatized by slab-derived fluids. Yet, whether similar metallogenic processes also operated in the Precambrian remains obscure. Here we address the issue by investigating the origin, $fO_2$, and S contents of calc-alkaline plutonic rocks associated with the Haib porphyry Cu deposit in the Paleoproterozoic Richtersveld Magmatic Arc (southern Namibia), an interpreted mature island-arc setting. We show that the ca. 1886–1881 Ma ore-forming magmas, originated from a mantle-dominated source with minor crustal contributions, were relatively oxidized (1–2 log units above the fayalite-magnetite-quartz redox buffer) and sulfur-rich. These results indicate that moderately oxidized, sulfur-rich arc magma associated with porphyry Cu mineralization already existed in the late Paleoproterozoic, probably as a result of recycling of sulfate-rich seawater or sediments from the subducted oceanic lithosphere at that time.

[1] Mineral Exploration Research Centre, Harquail School of Earth Sciences, Laurentian University, Sudbury, ON, Canada. [2] Department of Earth and Environmental Sciences, University of Michigan, Ann Arbor, MI, USA. [3] Geochronology and Tracers Facility, British Geological Survey, Nottingham, UK. [4] Department of Earth Sciences, University of Oxford, Oxford, UK. [5] DSI-NRF Centre of Excellence, University of Johannesburg, Johannesburg, South Africa. [6] School of Geosciences, University of Witwatersrand, Johannesburg, South Africa. [7] Canadian Centre for Isotopic Microanalysis, University of Alberta, Edmonton, AB, Canada. [8]Deceased: Jeremy P. Richards. ✉email: xmeng1@laurentian.ca

Porphyry Cu systems presently contribute ~75% of the world's Cu production and of the known deposits most formed in Phanerozoic arc-related settings[1,2]. The arc-related calc-alkaline, intermediate magmas responsible for most known porphyry Cu deposit formation are hydrous, moderately oxidized ($\Delta$FMQ + 1 to +2; where $\Delta$FMQ is $fO_2$ in log units relative to the fayalite-magnetite-quartz mineral redox buffer), and sulfur-rich, reflecting partial melting of the asthenospheric mantle metasomatized by slab-derived oxidized fluids[3–5]. Chalcophile metals (e.g., Cu) behave as incompatible elements in moderately oxidized magma and are transported to upper crustal levels where they partition efficiently into exsolving S- and Cl-bearing magmatic-hydrothermal ore fluids to form economic deposits under optimal ore-formation conditions[1,5].

The apparent rarity of porphyry Cu deposits in the Precambrian is attributed to poor preservation of upper crustal rocks in tectonically active environments, and/or unfavorable tectono-magmatic conditions which precluded their formation[6,7]. The predominance of ferrous iron and sulfide in terrigenous sediments, anoxic ocean water, and hydrothermally altered reduced submarine basalts during the Precambrian[8–10] suggest that a significant proportion of $H_2S$ would have been released to the sub-arc mantle in subduction zones, thus maintaining stability of mantle sulfides during partial melting in the source region of arc magmas[6,11,12], depleting juvenile silicate melts of chalcophile metals, and limiting the ore-forming potential of ascending magma[13,14].

The hypothesis that tectonomagmatic conditions in the Precambrian are unfavorable for porphyry Cu deposit formation remains poorly constrained because pervasive metamorphism, deformation, and alteration in Precambrian terrains have variably modified the primary mineralogy and textures of old rocks, rendering estimation of magmatic conditions and original composition difficult. Haib represents one of the largest and best-preserved Paleoproterozoic porphyry Cu deposits (indicated resource of 456.9 million tonnes at a Cu grade of 0.31%, using a cut-off grade of 0.25%, and minor Mo; https://www.deepsouthresources.com/) and only records minimal deformation and relatively low-grade metamorphism. Here we investigate the absolute timing of emplacement of the ore-related magmas at Haib using high-precision geochronology, and constrain the origin, and, in particular, the oxidation states and S contents of the magmas. Mineral inclusions in robust zircon and titanite were sought out to minimize potential effects of alteration and metamorphism. Contrary to previous knowledge[6,11], our findings demonstrate that similar metallogenic processes for Phanerozoic porphyry Cu deposits operated at ~1.88 Ga, a period of rapid crustal growth, oxygenation of Earth's atmosphere and oceans, and sulfur cycling in subduction zones.

## Results

**Geological setting**. The Haib deposit is hosted mainly by the high-K calc-alkaline intrusions of the Vioolsdrif Suite and subaerial volcanic rocks of the Orange River Group along the western part of the Paleoproterozoic Richtersveld Magmatic Arc (RMA) of southern Namibia[15–18] (Fig. 1, Supplementary Note 1 and Supplementary Figs. 1, 2), which formed in a mature island-arc setting at 1.85–1.91 Ga[15,18]. The pre-mineralization Orange River Group mainly contains plagioclase-phyric andesite porphyry and interleaved rhyolitic tuff (Fig. 1, Supplementary Fig. 2). The Vioolsdrif Suite mainly comprises equigranular granodiorite, with quartz-monzonite enclaves, and diorite succeeded by shallower syn-ore granodiorite porphyry and leucocratic granodiorite porphyry crosscut by poorly mineralized and carbonate-altered aplite

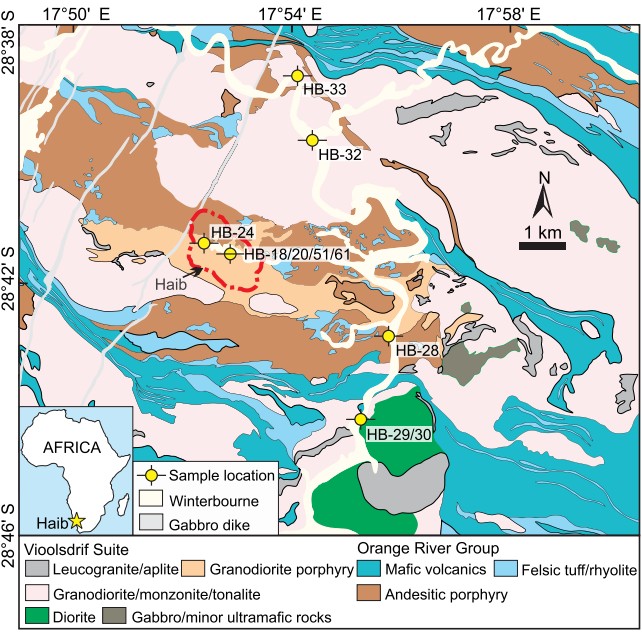

**Fig. 1 Simplified geological map of the Haib area showing main lithological units.** The map is modified from ref. [83]. The red dashed line represents the defined mineralization zone for Haib (report from Deep South Resources, Inc.).

dikes (Fig. 1, Supplementary Figs. 2 and 3). Rocks in the RMA, including the Haib deposit, evolved along a fractionation trend controlled by amphibole and minor plagioclase as indicated by listric-shaped rare-earth element (REE) patterns, moderate negative Eu anomalies[15], and a negative relationship between Dy/Yb ratios and $SiO_2$ contents (Fig. 2).

Disseminated and veinlet chalcopyrite mineralization with minor magnetite is associated with biotite ± K-feldspar ± epidote ± anhydrite ± titanite ± rutile alteration locally overprinted by chlorite and sericite alteration (Supplementary Figs. 1 and 4). Early dark micaceous veins are more common than A-, B-, and D-type veins (Supplementary Fig. 4; see terminology for the various vein types in ref. [19]), which reflects relatively deep levels of exposure of the system. Emplacement pressure for the plutonic host rocks and mineralization is estimated at ~200–300 MPa[16], which is comparable to deeper parts of typical Phanerozoic porphyry Cu systems. The area underwent metamorphism up to greenschist facies and localized intense shearing in the volcanic rocks during the ~1.1 Ga Namaqua orogeny[20].

**High-precision U–Pb geochronology**. Zircon U–Pb geochronology by laser ablation multi-collector inductively coupled plasma-mass spectrometry (LA-MC-ICP-MS) and chemical abrasion isotope dilution thermal ionization mass spectrometry (CA-ID-TIMS) constrains the absolute timing of the crystallization of the igneous rocks and related mineralization. Zircons from two samples of andesite porphyry and one sample of rhyolitic tuff yielded LA-MC-ICP-MS upper concordia intercept ages of 1912 ± 10 Ma, 1892 ± 9 Ma, and 1892 ± 6 Ma, respectively (Fig. 3a, Supplementary Fig. 5). Zircon from equigranular granodiorite, quartz-monzonite enclave, and diorite of the Vioolsdrif Suite yielded LA-MC-ICP-MS upper concordia intercept ages of 1887 ± 7 Ma, 1886 ± 5 Ma, and 1875 ± 10 Ma, respectively (Fig. 3a, Supplementary Fig. 5). These ages are consistent with ages for other rocks in the RMA[15,17].

Mitigation of Pb loss in zircons from samples of mineralized granodiorite porphyry and leucocratic granodiorite porphyry, and

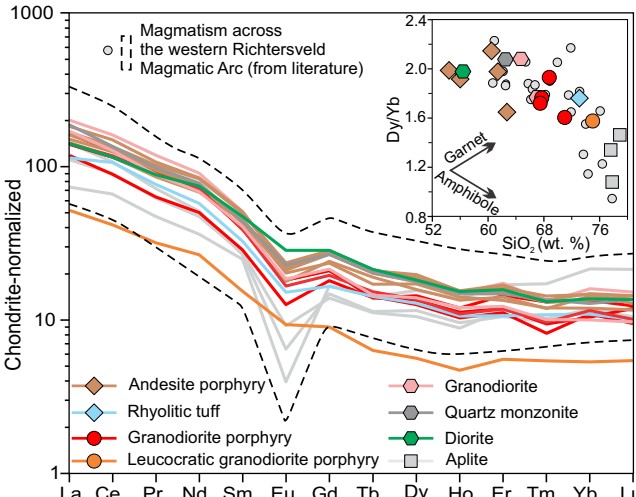

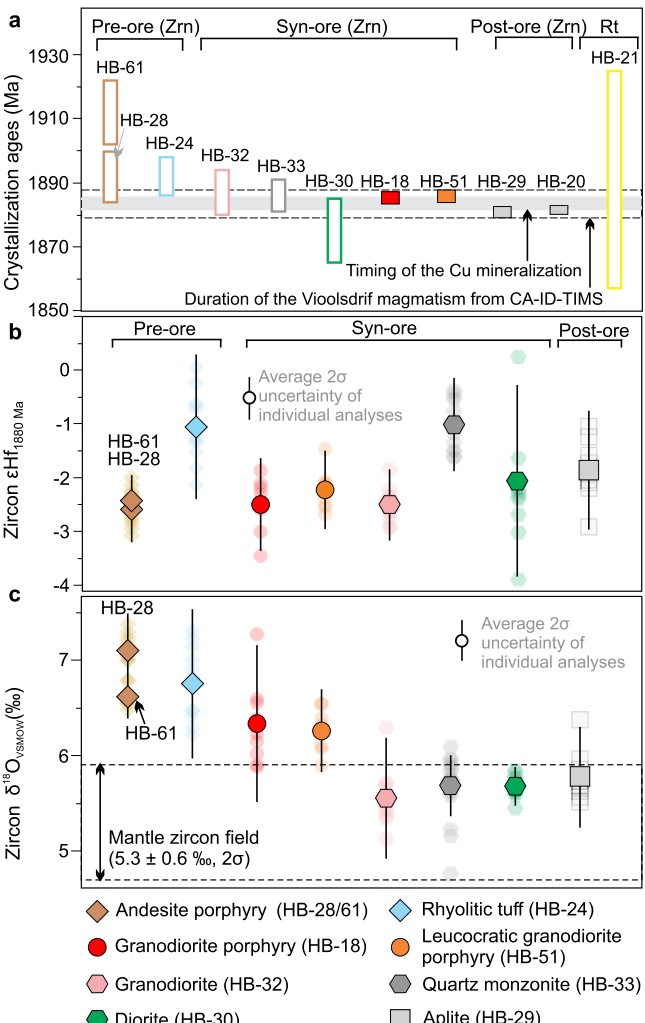

**Fig. 2 Chondrite-normalized rare-earth element spider diagram for igneous rocks from Haib.** The normalization values are from ref. [84]. Inset is Dy/Yb versus $SiO_2$ diagram, and the fractionation trends of amphibole and garnet are from ref. [42]. Data for the igneous rocks (LOI ≤ 2 wt.%) from the western Richtersveld Magmatic Arc were compiled in ref. [15].

poorly mineralized aplite was accomplished using chemical abrasion, and the CA-ID-TIMS U–Pb isotopic system provides the most precise dates (Fig. 3a, Supplementary Fig. 6). These three samples yielded weighted mean $^{207}Pb/^{206}Pb$ ages (Th-corrected) of 1885.47 ± 0.93, 1886.0 ± 1.6, and 1881.02 ± 0.71 Ma, respectively (Fig. 3a, Supplementary Fig. 6). Mild chemical abrasion of six small zircon grains (<30 μm) from one carbonate-altered aplite dike from the ore zone partially mitigated Pb loss, with five of the six grains yielding a weighted mean $^{207}Pb/^{206}Pb$ age (Th-corrected) of 1881.8 ± 1.2 Ma (Fig. 3a, Supplementary Fig. 6). These results are interpreted to best constrain the emplacement age of the Vioolsdrif Suite and are consistent with the LA-ICP-MS zircon U–Pb age data (Fig. 3a). The age range between the granodiorite porphyry and the carbonate-altered aplite brackets the porphyry Cu mineralization to 1883.6 ± 1.8 Ma, which broadly agrees, within uncertainty, with the LA-ICP-MS upper concordia intercept U–Pb age of 1891 ± 34 Ma for hydrothermal rutile associated with chalcopyrite mineralization (Fig. 3a, Supplementary Fig. 7).

**Zircon Hf-O isotopes.** Xenocrysts are rare in zircons from the igneous rocks sampled. Some petrographically distinct cores identified as antecrysts yielded LA-ICP-MS $^{207}Pb/^{206}Pb$ dates that are indistinguishable from ages of the host rocks (Supplementary Fig. 5). Zircons from these rocks yielded initial $\varepsilon_{Hf}$ values of −2.6 ± 0.6 to −1.0 ± 0.9 (2σ, n = 99, −2.0 ± 1.6 on average) and $\delta^{18}O$ values of 5.56 ± 0.64 ‰ to 7.10 ± 0.39 ‰ (2σ, n = 102, 6.14 ± 1.19 ‰ on average; Fig. 3b, c). Considering the rarity of zircon inheritance, the narrow ranges of the isotope ratios for the igneous rocks are consistent with homogenized primary arc magma variably contaminated with recycled upper crustal materials (e.g., from an earlier arc nearby RMA[15]) in its source region. The elevated zircon $\delta^{18}O$ values for the volcanic and porphyritic rocks compared to the equigranular rocks (Fig. 3c) indicate more contamination from the recycled upper crustal materials that had undergone low-temperature alteration[21]. The near-constant Hf isotope ratios among the samples (Fig. 3b) suggest a limited effect of the crustal contaminants on the Hf isotopic systematics of the source and also support insignificant crustal assimilation during magma ascent.

**Fig. 3 Zircon U–Pb-Hf-O and rutile U–Pb isotopes for representative rocks in Haib. a** Crystallization ages of magmatic zircons from ten pre-, syn-, and post-ore igneous rocks (identified from cross-cutting relationship, details in Supplementary Note 1) and a hydrothermal rutile (sample HB-21, syn-ore granodiorite porphyry) associated with chalcopyrite mineralization in Haib, dated using LA-ICP-MS (samples HB-61, HB-28, HB-24, HB-32, HB-33, and HB-30) and CA-ID-TIMS (samples HB-18, HB-51, HB-29, and HB-20) methods. **b** Zircon Hf isotopic composition (as $\varepsilon_{Hf}$ calculated at 1880 Ma). **c** Zircon $\delta^{18}O$ ratios for representative igneous samples. Filled symbols in (**b**) and (**c**) are average values for each sample; smaller pale symbols represent individual analyses. Mantle zircon field in (**c**) is from ref. [85]. Error bars indicate 2σ uncertainties. See Supplementary Data 1 for sample locations and descriptions.

**Relatively oxidized magmas.** The $fO_2$ of the causative Haib magmas, from which the mineralizing ore fluid evolved, is estimated using three independent oxybarometers (see Methods in refs. [22–24], Fig. 4a–c). Magmatic $fO_2$ is estimated using synchrotron-based micro X-ray absorption near-edge structure spectroscopy (μ-XANES) at the S $K$-edge to measure the $S^{6+}/\Sigma S$ ratios of zircon-hosted F-rich apatite inclusions remote from fractures (Fig. 4a, Supplementary Fig. 8). Because most zircons yielded concordant U–Pb dates indicating closed system behavior (Supplementary Figs. 5 and 6), analysis of apatite inclusions in the zircons eliminates the effects of post-crystallization metamorphism and/or alteration. The equant to sub-equant morphology of these monophase apatite inclusions and the absence of other coeval silicate mineral inclusions are consistent with

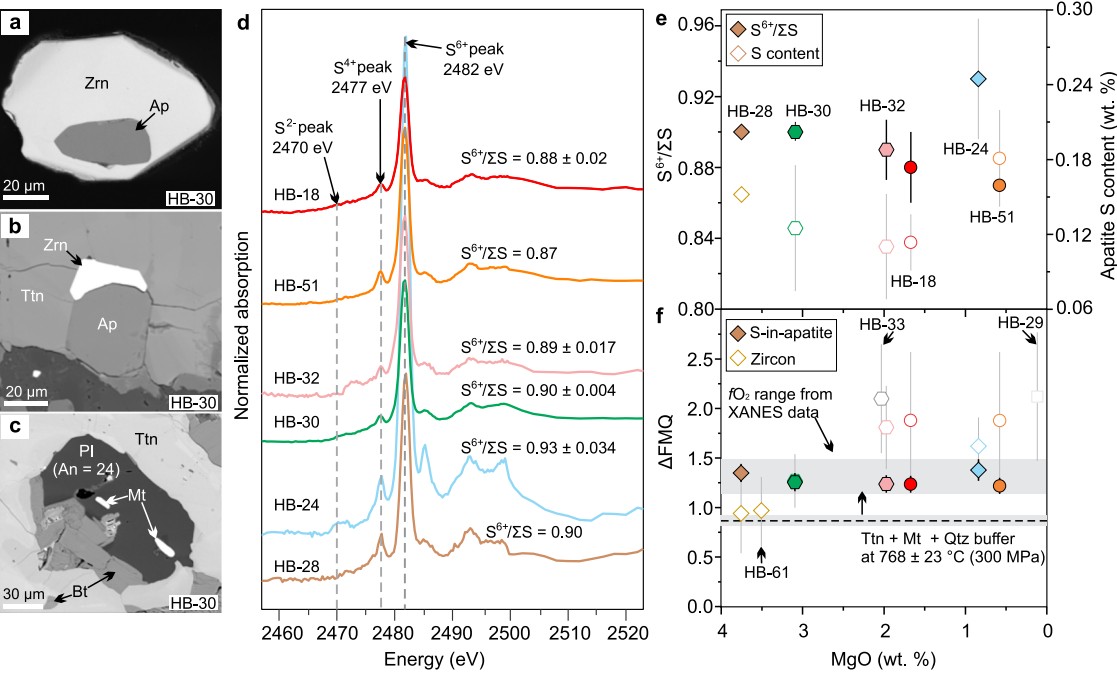

**Fig. 4 Backscattered electron images of inclusions in titanite and zircon, apatite μ-XANES spectra and sulfur contents, and results of magmatic $fO_2$ estimation. a** Representative apatite inclusion in zircon from sample HB-30; BSE. **b** Zircon overgrowth on apatite with both hosted in magmatic titanite from sample HB-30; BSE. **c** Biotite, plagioclase (An number = 24, calculated from energy-dispersive X-ray analysis), and magnetite in titanite from diorite sample HB-30; BSE. **d** Normalized μ-XANES spectra at the S K-edge of apatite in zircons for representative igneous rocks from Haib. Spectra for only one apatite inclusion (n = 1) were obtained for samples HB-28 and HB-51, respectively, while the others represent averaged spectra of a number of apatite inclusions (HB-18, n = 4; HB-24, n = 2; HB-30, n = 3; HB-32, n = 3). Peak positions for $S^{2-}$, $S^{4+}$, and $S^{6+}$ are at 2470 eV, 2477 eV, and 2482 eV, respectively, and are shown as dotted gray lines. The average $S^{6+}/\Sigma S$ ratios (1σ) are calculated from the merged spectra following ref. [22]. **e** Plot of apatite $S^{6+}/\Sigma S$ ratios and S contents versus MgO for the representative samples. **f** ΔFMQ values versus MgO for the representative samples. Error bars in (**e**) and (**f**) represent 1σ uncertainties. An anorthite, Ap apatite, Bt biotite, Mt magnetite, Pl plagioclase, Ttn titanite, Zrn zircon. Symbols and colors are the same as in Fig. 3.

crystallization as early mineral phases under near-equilibrium, near-liquidus conditions (Fig. 4a, b). Based on thermodynamic models in ref. [25] that predict apatite halogen compositions evolve along different trajectories during volatile-undersaturated and water-saturated crystallization, these fluorapatite inclusions are interpreted to have crystallized prior to the melt reaching volatile saturation (Supplementary Note 2, Supplementary Fig. 9). The μ-XANES data reveal that sulfur in apatite is dominantly $S^{6+}$ (Fig. 4d, Supplementary Fig. 10), and integrated $S^{6+}/\Sigma S$ peak area ratios (Fig. 4e) yield $fO_2$ values that range from ΔFMQ + 0.85 ± 0.08 to ΔFMQ + 1.07 ± 0.11 (1σ, ΔFMQ + 0.89 ± 0.04 on average) based on the apatite oxybarometer experimentally calibrated at 1000 °C and 300 MPa[22]. Changes in temperature and pressure are reported to shift the sulfide–sulfate transition in $fO_2$ space for silicate glasses[26,27]. Thus if we consider that the effect of temperature and pressure on silicate glasses is proportional to the effect of sulfur incorporation into apatite[22], the P-T-corrected $fO_2$ values for the Haib plutonic and volcanic rocks are ΔFMQ + 1.24 ± 0.01 and ΔFMQ + 1.37 ± 0.02 (1σ) respectively (Fig. 4f; Supplementary Note 3). The results for the plutonic rocks are interpreted to provide the best estimate of the pre-degassed redox state of the ore-forming magmas.

The Ce, Ti, and age-corrected initial U concentrations in zircon from the syn-ore plutonic rocks (the diorite, granodiorite and quartz monzonite, granodiorite porphyries) yield magmatic $fO_2$ values ranging from ΔFMQ + 1.3 ± 0.3 to ΔFMQ + 2.1 ± 0.6 (1σ, ΔFMQ + 1.9 ± 0.6 on average) using a new empirical oxybarometer[23] (see screening criteria of zircon geochemistry data in notes in Supplementary Data 5). These results are consistent with the titanite + magnetite + quartz assemblage in the diorite which constrains the redox state of the melt to be >ΔFMQ + 0.7 ± 0.1 at 768 ± 23 °C (1σ, temperature calculated using titanium-in-zircon thermometer)[24,28] (Fig. 2c, f), although the result is interpreted to be less reliable because of the difficulty in finding credible equilibrated mineral assemblages.

**Sulfur-rich apatite and melt.** Magmatic apatite inclusions in zircon and titanite from the Haib system contain relatively high S contents ranging from 0.11 ± 0.04 to 0.19 ± 0.08 wt.% (Fig. 4e, Supplementary Data 9), which are comparable to those reported for basaltic-andesitic to rhyolitic arc magmas and igneous rocks associated with Phanerozoic porphyry Cu systems (Fig. 5a). Recent studies suggest that the partition coefficient for S between apatite and silicate melt ($D_S^{ap/m}$) varies with magmatic $fO_2$ and temperature and is not significantly affected by melt composition[22,29,30]. Because a $D_S^{ap/m}$ model involving all these parameters is not currently available, we developed a method based on published experimental results (Method 1) and used a published thermodynamic method (Method 2) to determine partition coefficients for S between apatite and silicate melt (see details in 'Methods' section), which in turn were used to estimate melt S contents for the Haib samples that range from 0.07 ± 0.03 to 0.17 ± 0.08 wt.% S (1σ, 0.11 ± 0.06 wt.% on average), and 0.02 ± 0.001 to 0.08 ± 0.04 wt.% S (1σ, 0.04 ± 0.03 wt.% on average), respectively. The two independent estimates of melt S content agree within one order of magnitude and are broadly similar to S concentrations in many Phanerozoic arc melts (SiO₂ ≥ 52 wt.%, with limited degassing; Fig. 5b) and those associated with Phanerozoic porphyry Cu systems.

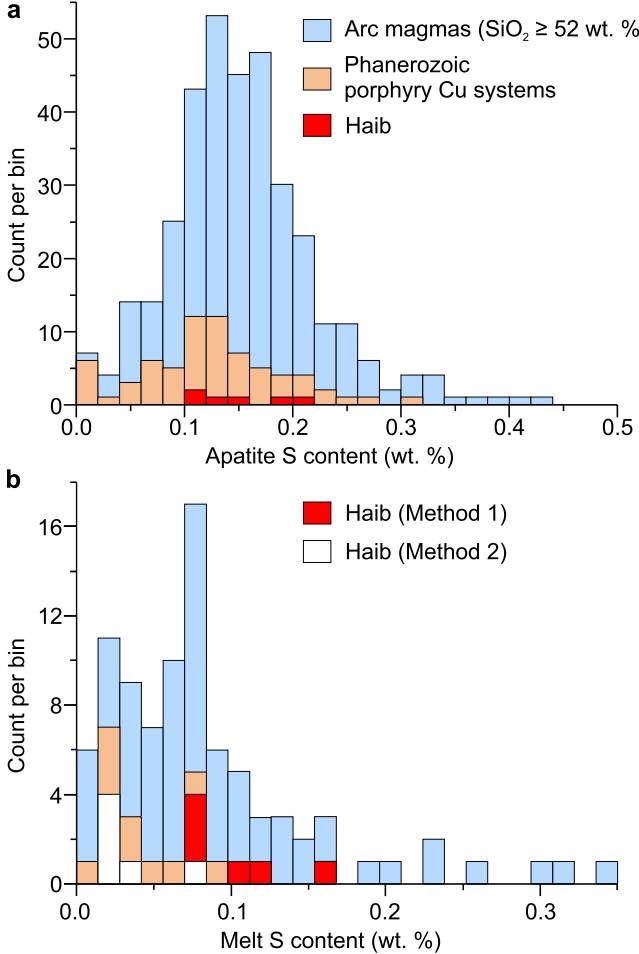

**Fig. 5 Histograms of S concentrations of igneous apatites and melt inclusions from arc magmas, rocks associated with Phanerozoic porphyry Cu systems and the Haib. a** Igneous apatites from intermediate to felsic arc magmas (0.15 ± 0.07 wt.% S on average; 1σ, $n = 349$, basaltic-andesitic to rhyolitic composition with $SiO_2$ presumably ≥52 wt.%; most of them are from the Circum-Pacific volcanic arcs) and a subset from those associated with Phanerozoic porphyry Cu systems (0.12 ± 0.065 wt.% S on average; 1σ, $n = 64$) and Haib ($n = 6$). **b** Measured S concentrations of melt inclusions in olivine and minor pyroxene in arc rocks (0.09 ± 0.07 wt.% on average; 1σ, $n = 69$; $SiO_2$ ≥ 52 wt.%; data from GEOROC database) and in rocks associated with Phanerozoic porphyry Cu systems, as well as those estimated using apatite-melt partition coefficients for S from the Haib igneous apatites. The melt inclusion data for arc rocks were filtered using $CO_2$ ≥ 50 ppm to identify those with limited degassing (see ref. [86]). N represents numbers of samples or groups of apatites and melt inclusions. See data sources in Supplementary Note 4.

## Discussion

The calc-alkaline arc-type magmas, from which the mineralizing fluid evolved, formed at ~1886–1881 Ma, coinciding with a period of rapid crustal growth[31] and supercontinent assembly in Earth history. Pervasive hydrothermal anhydrite, part of the potassic alteration stage, is associated with chalcopyrite mineralization in the Haib deposit and indicates the hydrothermal system was relatively oxidized and consistent with exsolution of hydrothermal fluids from a relatively oxidized ore-related magma. The pre-degassed magmas were moderately oxidized (ΔFMQ + 1–2) and sulfur-rich, which are salient characteristics for Phanerozoic arc magmatism.

The elevated magmatic $fO_2$ values of the Haib magmas relative to mid-oceanic ridge basalts (MORB; ΔFMQ − 2–0)[32] are strongly reminiscent of Phanerozoic oxidized island-arc magmas that are generally thought to originate from the oxidized sub-arc mantle[3,4,11,12,33–38]. Deep crustal garnet fractionation has recently been hypothesized to auto-oxidize magma for porphyry Cu deposit formation[39,40]. However, the Haib rocks have listric REE patterns and moderately negative Eu anomalies which indicate low-pressure differentiation with amphibole ± plagioclase fractionation (Fig. 2). Amphibole preferentially incorporates middle REEs over heavy REEs and garnet incorporates heavy REEs relative to light REEs, thus amphibole fractionation will decrease, but garnet fractionation will increase, residue melts of Dy/Yb ratios[41,42]. The decreasing Dy/Yb ratios with increasing $SiO_2$ content (Fig. 2) support amphibole-dominated fractionation without or with minimal garnet influence, leading us to discount the possibility that the elevated magmatic $fO_2$ of the Haib magmas is a result of garnet fractionation. In addition, the narrow ranges of zircon Hf and O isotopic compositions combined with the rarity of zircon inheritance (i.e., xenocrysts) in the Haib plutonic rocks suggest that modification of the magmatic $fO_2$ by crustal contamination during magma ascent is insignificant.

Considering that modern plate tectonics or its equivalent may have, at least on a local scale, operated since 1.8–1.9 Ga or earlier[43–46], we argue that the Haib magmas are more likely derived from partial melting of the mantle lithosphere that was modified by oxidized slab-derived fluids in the Paleoproterozoic. This is consistent with the recent findings that thick layers of evaporites and an abrupt elevation of marine sulfate content (up to at least 10 mmol/kg, compared to modern seawater sulfate content of 28 mmol/kg) occurred at ~2.0 Ga[47–50], which supports the hypothesis that surface sulfate would have been available to be recycled into the mantle since the middle Paleoproterozoic. Although the hydrothermally altered oceanic basalt at that time is considered to be relatively reduced[9], the sulfate-rich fluid released from the oxidized sediments in the subducted oceanic lithosphere may have been able to oxidize the sub-arc mantle without direct addition of $Fe^{3+}$[3,51], culminating in the formation of relatively oxidized and sulfur-rich magmas.

The oxidized and sulfur-rich features of the derivative basaltic to dacitic magma can delay high-volume sulfide saturation and deep-crustal loss of chalcophile metals, which is the first-order control required for the magma to form the preserved Paleoproterozoic porphyry Cu deposit upon its emplacement in the upper crust[52]. Our findings suggest that similar metallogenic processes for Phanerozoic porphyry Cu deposits have operated much earlier than generally considered, as recently suggested by others[6,11].

## Methods

**Sample preparation.** Sixty-two samples of volcanic and intrusive igneous rocks, including a minority of altered and vein-type samples, were collected from drill core and outcrops in October 2018. To investigate the nature of the magmatism, least-altered samples were targeted for collection, but most of the drill core samples are variably altered owing to proximity to the ore deposit. Nineteen of the petrographically least-altered samples were chosen for analytical work (Supplementary Data 1).

Zircons were separated from representative samples of the major igneous phases by traditional magnetic and density methods after electric-pulse disaggregation at Overburden Drilling Management Inc., Ottawa, Canada. Zircon grains devoid of cracks and mineral and/or fluid inclusions as well as inclusion-rich grains were hand-picked and mounted in separate sets of epoxy pucks and polished to expose their middle-sections. Magmatic zircon and titanite crystals, and contained mineral inclusions, as well as hydrothermal rutile, were characterized using a Tescan Vega 3 scanning electron microscope with backscattered electron (BSE) imaging, cathodoluminescence, and Bruker energy-dispersive spectroscopy (EDS) at Laurentian University.

**Whole-rock geochemistry.** Eighteen of the least-altered samples collected from drill cores and outcrops from Haib were crushed and ground at Australian Laboratory Services (ALS), Sudbury, Canada for whole-rock geochemical analyses.

The powdered samples were analyzed using X-ray fluorescence (XRF) and inductively coupled plasma-mass spectrometry (ICP-MS) with pre-fusion (lithium metaborate; CCP-PKG01 analytical package) at ALS, Vancouver, Canada. One secondary reference sample LK-NIP-1 (Nipissing diabase, https://www.mndm.gov.on.ca/sites/default/files/2018_geo_labs_brochure.pdf) was analyzed as an unknown to verify the analytical quality during the analytical session, and the results are consistent with standard values (see Supplementary Data 2). Some elements of samples HB-23, HB-51, and HB-61 were duplicated. The results are reported in Supplementary Data 2.

**Zircon U–Pb isotope and trace element analyses**. Zircon uranium–lead (U–Pb) isotope and trace element analyses were conducted in the Mineral Exploration Research Centre—Isotope Geochemistry Laboratory (MERC-IGL) at Laurentian University. A 193 nm Photon Machines Analyte G2 ArF excimer laser equipped with a two-volume Helex II laser cell was used to ablate the zircons. To minimize the common lead contamination from surface material prior to ablation, the samples were polished using alumina powder, put in an ultrasonic bath with milli-Q water, and cleaned with ethanol by Kim wipe. The detailed instrumental parameters of laser ablation and ICP-MS, as well as data processing, for these analyses are listed in Supplementary Data 3.

The parameters for the laser were a fluence of 2 J/cm², a repetition rate of 7 Hz, and a spot size of 35 μm. Ablation duration was 10 s, leaving an estimated ablation pit depth of <5 μm. The ablated aerosol downstream of the sample cell was split into two mass spectrometers for measuring the U–Pb isotope ratios and trace element abundances simultaneously[53]. U–Pb isotopic ratios, and U, Th, and Pb concentrations were measured on a Thermo Scientific Neptune Plus multi-collector ICP-MS equipped with a Jet interface and nine Faraday cups, whereas trace elements were analyzed on a Thermo Scientific iCap triple-quadrupole (TQ) ICP-MS in single quad mode to ensure maximum sensitivity on low to intermediate mass range. Sixty seconds of background were measured at the beginning and end of each analytical session, with 30 s of background measured between each analysis.

Correction of laser-induced element fractionation, instrumental drift, and downhole fractionation was performed with the U–Pb Geochronology data reduction scheme implemented within Iolite v. 3.6 software[54,55], with U–Pb isotope ratios normalized to the zircon primary reference material OGC-01 that was periodically dispersed into the analytical sessions. The secondary reference materials TanBrA and Grn were analyzed to monitor the accuracy and reproducibility of the unknown analyses, with results consistent with standard values (Supplementary Data 4). Within-run variance in the measured ratios for OGC-01 was propagated into the 2SE uncertainty for all unknowns. No additional uncertainty propagation was applied to the $^{207}Pb/^{206}Pb$ ratios of the unknown analyses because long-term variance of $^{207}Pb/^{206}Pb$ ratios of the verification reference materials across all sessions is limited with statically acceptable MSWD (mean square of weighted deviates; that is, ~1). Concordia age calculation and data plotting were completed using Isoplot v. 4.15[56], in which uranium decay constant uncertainties were not considered. The MSWD of the dates are statistically acceptable for population sizes at the 95% confidence interval. Trace element data were processed using the internal standard data reduction scheme within Iolite v. 3.6 and normalized to the synthetic glass NIST610. The stoichiometric concentration of Si in zircon was assumed to be 15.284 wt.%. The synthetic glass NIST612 and zircon 91500 were used to verify the accuracy of the analyses. The results are reported in Supplementary Data 4.

**CA-ID-TIMS zircon U–Pb dating**. Chemical abrasion isotope dilution thermal ionization mass spectrometry (CA-ID-TIMS) U–Pb geochronology of single zircon crystals and fragments was undertaken at the Geochronology and Tracers Facility, British Geological Survey, Keyworth, the UK. Chemical abrasion was undertaken at 190 °C for 12 h for samples HB-18, HB-29, and HB-51, within the exception of sample HB-20 that was leached at 180 °C following the method in ref. [57]. The methodology for all other analytical procedures, instrumental conditions, corrections, and data reduction follows that outlined in detail in ref. [58] using the ET2535 tracer[59,60]. Isotope ratio measurements were made using a Thermo Triton thermal ionization mass-spectrometer (TIMS), with the U decay constants of ref. [61], the $^{238}U/^{235}U$ ratio of ref. [62], and the $^{230}Th$ decay constants of ref. [63]. The $^{206}Pb/^{238}U$ and $^{207}Pb/^{206}Pb$ dates were corrected for initial $^{230}Th$ disequilibrium[64] using a value of Th/U [magma] = 3.5. The results are reported in Supplementary Data 5.

The best estimate of emplacement age was selected from the population of concordant data where the $^{207}Pb/^{206}Pb$ weighted mean date had a statistically acceptable MSWD for the given population size at 95% confidence interval. For zircons from sample HB-20, the Pb loss has not been completely mitigated, we report the weighted mean $^{207}Pb/^{206}Pb$ age with acceptable MSWD of 0.15. All uncertainties are reported at 2σ level considering analytical and tracer calibration uncertainty but excluding the contribution from uncertainties of decay constants.

**Rutile U–Pb isotopes**. In-situ U–Pb isotopes of rutile in thin section were measured using the Thermo Neptune Plus MC-ICP-MS coupled to a 193 nm Photon Machines Analyte G2 laser equipped with a two-volume Helex II laser cell in MERC-IGL at Laurentian University. The laser ablation was carried out with a fluence of 2–3 J·cm⁻², a repetition rate of 7 Hz, and a spot size of 35 μm. Each

analysis lasted for 35 s. Primary reference material was R10 (1095 ± 4.7 Ma)[65] and secondary materials were R19 (489.5 ± 0.9 Ma)[66] and Sugluk (1723 ± 6.8 Ma)[67], which were periodically analyzed throughout the analytical run. The detailed instrumental parameters of laser ablation and ICP-MS, as well as data processing, are listed in Supplementary Data 3. The results for secondary reference materials of rutile are reported in Supplementary Data 6 and are in agreement with the published standard values. Results for unknown analyses are also reported in Supplementary Data 6.

**Zircon Lu–Hf isotopes**. Lu–Hf isotopes were measured on the same spots as O isotopes and some of the zircon U–Pb isotope and trace element analyses. The same analytical equipment as for the U–Pb analyses were used. Laser ablation was carried out with a fluence of 6 J cm⁻², a repetition rate of 7 Hz, and a spot size of 40 μm. Ablation duration was 60 s, leaving an estimated ablation pit depth of ~30 μm. One primary and five secondary reference standards (Pleosovice; 91500, R33, FC1, OGC, MUN1, and MUN3) were periodically analyzed during the analytical runs. The details for the instrumental parameters of laser ablation and ICP-MS, as well as data processing, for these analyses are listed in Supplementary Data 3.

The raw Hf isotope data were processed in Iolite v.3.6, with baseline subtraction, instrumental drift, and mass bias corrections performed with a modified version of the Hf isotope data reduction scheme. Within-run variance in the measured ratios for Plesovice was propagated into the 2SE uncertainty for all unknowns. Similar to the U–Pb isotopic analyses, no additional uncertainty propagation was applied to the Hf isotope ratios for the unknown analyses. Instrumental mass bias and interference correction factors were determined within the session through the iterative calculation of the effective $^{176}Yb/^{173}Yb$ ratio required to yield identical $^{176}Hf/^{177}Hf$ ratios of MUN1 and MUN3. The effective (mass bias-corrected) within-session $^{176}Yb/^{173}Yb$ was then applied in the interference correction for all analyses as part of the data reduction scheme. The corrected $^{176}Hf/^{177}Hf$ ratios are 0.282309 ± 0.000030 (2σ, $n = 13$) for 91500, 0.282750 ± 0.000022 (2σ, $n = 13$) for R33, 0.282184 ± 0.000031 (2σ, $n = 13$) for FC1, 0.280558 ± 0.000054 (2σ, $n = 13$) for OGC, and 0.282127 ± 0.000019 (2σ, $n = 10$) for MUN. These results are identical to the published values within uncertainties[68–70]. The initial $^{176}Hf/^{177}Hf$ ratios and $\varepsilon_{Hf}(t)$ values are calculated using a $^{176}Lu$ decay constant of $1.867 \times 10^{-11}/a$[71] and the chondritic parameters from ref. [72]. The results are reported in Supplementary Data 7.

**Zircon O isotopes**. Oxygen isotopes ($^{18}O$ and $^{16}O$) in zircon were analyzed using a Cameca IMS 1280 multi-collector ion microprobe in the Canadian Centre for Isotopic Microanalysis at the University of Alberta. Most of the samples were analyzed in session IP19029 prior to U–Pb–Lu–Hf isotope analyses, except for samples HB-30 and HB-33 which were analyzed in session IP20013 and the analyses were conducted on different sets of grains for U–Pb-Lu–Hf isotopes. Counts of $^{16}O^1H/^{16}O$ for zircons in the session IP19029, in which some of the samples were collected from the ore deposit, were simultaneously analyzed with the O isotopes to monitor the effect of hydrothermal alteration. Most of the analyses yielded low $^{16}O^1H/^{16}O$, and only a few analyses having significantly higher values than the standards (Delta OH > 40%) were excluded.

The zircons were analyzed using a 20 keV $^{133}Cs+$ primary beam with a spot size of ~10 μm. Detailed instrumental parameters have been described by ref. [73]. Instrumental mass fractionation was monitored by repeated analysis of the zircon primary reference material (S0081, $\delta^{18}O_{VSMOW} = + 4.87$; R. Stern, unpublished laser fluorination data, University of Oregon) after every four unknown analyses. The $^{18}O-/^{16}O-$ dataset for S0081 was processed collectively for two analytical sessions ($n = 64$ and 12), yielding standard deviations of 0.12‰ and 0.09‰, respectively, following correction for a systematic within-session drift of ≤0.4‰. Zircon secondary reference material TEM2 was analyzed following every eight unknown analyses. Multiple spots on different TEM2 grains yielded weighted mean $\delta^{18}O_{VSMOW}$ values of +8.23 ± 0.12 ‰ (SD, $n = 42$, MSWD = 1.07) for session IP19029 and +8.27 ± 0.09‰ (SD, $n = 23$, MSWD = 0.74) for session IP20013, which is consistent with the published standard $\delta^{18}O_{VSMOW}$ value of +8.2‰[74]. The individual spot uncertainties of $\delta^{18}O_{VSMOW}$ for unknowns in the two sessions have medians of ±0.25‰ and ±0.20‰ at 95% confidence, which includes errors relating to within-spot counting statistics, geometric effects, and correction for instrumental mass fractionation. The results are reported in Supplementary Data 8.

**Electron microprobe analysis**. Major and minor element compositions of apatite were acquired using a Cameca SX100 electron microprobe analyzer with wavelength-dispersive spectroscopy at Ontario Geosciences Laboratory, with results reported in Supplementary Data 9. An accelerating voltage of 15 keV, a beam current of 10 nA, and rastered beam sizes of 2 or 5 μm (depending on the apatite grain size) were used for all element analyses. X-ray lines, analyzing crystals, and counting times (for both peak and background measurements) were as follows: P $K\alpha$, LPET5, 15 s; Si $K\alpha$, LTAP2, 20 s; Al $K\alpha$, LTAP2, 20 s; Mg $K\alpha$, LTAP2, 20 s; Ca $K\alpha$, LPET5, 15 s; Mn $K\alpha$, LiF4, 40 s; Fe $K\alpha$, LiF4, 40 s; Sr $L\alpha$, LPET5, 30 s; Na $K\alpha$, LTAP2, 20 s; K $K\alpha$, PET3, 20 s; S $K\alpha$, LPET3, 60 s; F $K\alpha$, PCO, 20 s; Cl $K\alpha$, LPET5, 20 s; Zr $L\alpha$, LPET5, 5 s. The detection limits for these elements were calculated to be as follows: $P_2O_5$, 680 ppm; $SiO_2$, 130 ppm; $Al_2O_3$, 140 ppm; MgO, 140 ppm; CaO, 340 ppm; MnO, 520 ppm; FeO, 520 ppm; SrO, 850 ppm; $Na_2O$, 170 ppm; $K_2O$, 140 ppm; $SO_3$, 270 ppm; F, 880 ppm; and Cl, 190 ppm.

Due to the small size of the apatites analyzed, zirconium concentrations were measured to monitor the contamination from the host zircon crystal. Analyses with $ZrO_2 > 1\%$ were excluded. The decomposition of apatite caused by electron beam damage[75,76] under the conditions used for this study was also tested (see Supplementary Data 9). The compositions of the test samples (beam size of 2 μm) typically show, with analytical time: (1) an increase or decrease in F count rate; (2) slightly decrease in Cl count rate; and (3) a nearly constant S count rate. The test demonstrates that the S and Cl abundances are reliable and the F concentrations might not be well constrained. Beam damage of apatite obtained at a larger beam size of 5 μm can be minimized, particularly for apatite grains with c-axis perpendicular to the electron beam.

**Micro X-ray absorption near-edge structure spectroscopy (μ-XANES)**. To measure the sulfur oxidation states of the pristine apatite inclusions (i.e., previously not analyzed by electron microprobe), micro X-ray absorption near-edge structure spectroscopy at the sulfur K-edge (S μ-XANES) was conducted at the Advanced Photon Source (APS) of the Argonne National Laboratory in Chicago, IL, USA, and at the Swiss Light Source (SLS) of the Paul Scherrer Institute in Villigen, Switzerland. The analyses at the APS were performed at the GSECARS X-ray microprobe beamline on sector 13-ID-E and those at the SLS were performed at the PHOENIX X07MA/B tender X-ray microspectroscopy beamline. The APS beam-line has an energy range of 2.3–28 keV, whereas the SLS beamline has an energy range of 0.35–8 keV. Both beamlines use undulator X-ray sources to provide photons for μ-XANES measurements and employ double crystal (silicon 111) monochromators to generate monochromatic radiation focused with a Kirkpatrick–Baez mirror system. The monochromatic beam was focused down to a size of $2 \times 2$ μm at APS and 3 μm diameter at SLS. The energy at both beamlines was calibrated to the ~2482 eV white line of sulfate, using clear double-sided sticky tape at APS (2481.8 eV) and powdered $CaSO_4$ at SLS (2482 eV). Differences in the calibration are attributed to slight differences in monochromator calibration and setup.

To minimize the Compton and Rayleigh scattering contribution, the polished surface of the samples was positioned at 45° from the incoming beam. We followed the collection and processing procedures described in refs. [22,77]. XRF maps were acquired at the beamlines immediately proceeding spectra collection to locate the apatite inclusions in zircon. The incoming energy for maps was fixed at 2482 eV and measured fluorescence lines of S, Si, P, and Al to discriminate different phases in the maps (e.g., zircon, apatite, and epoxy). The dwell time of the detector was set at 1 s for each $3 \times 3$ μm pixel at SLS and 0.02 s for each $2 \times 2$ μm pixel at APS. At both beamlines sulfur fluorescence spectra were collected using four-element silicon drift diode detector arrays positioned at an angle of 90° to the incident beam and corrected for detector dead time.

The S μ-XANES spectra were collected by scanning the incident beam energy from 2440 to 2550 eV using a slightly different setup at each beamline, but all scans were divided into the same three energy regions: (i) a pre-edge region from 2440–2460 eV, (ii) the edge region from 2460–2500 eV, and (iii) a post-edge region from 2500–2550 eV. At APS, step sizes of 1, 0.3, and 2 eV were used in the pre-edge, edge, and post-edge regions, respectively, while at SLS step sizes of 2, 0.3, and 2 eV were used. The counting duration for all regions at APS was 3 s and at SLS was 1 s per step in the pre- and post-edge regions and 2 s per edge step. The total scan duration was ~9 min at APS and ~12 min at SLS, with differences in beamline mechanics and scan setup resulting in different scan durations. The European Synchrotron Radiation Facility S K-edge XANES spectra database was used to identify the $S^{6+}$ (~2482 eV; anhydrite), $S^{4+}$ (~2478 eV; sodium sulfite), and $S^{2-}$ (~2470 eV; pyrrhotite) peak energy position for the unknowns. Spectra of pyrrhotite ($Fe_{1-x}S$) and anhydrite powder ($CaSO_4$) are shown in Supplementary Fig. 10a to illustrate the differences in $S^{2-}$ from $S^{6+}$ spectra.

Two scans were collected at each point to improve counting statistics and monitor for beam damage caused by irradiation of the focused X-ray beam and contribution from other phases (e.g., host zircon, epoxy). Beam damage manifests as the growth or shrinkage of absorption peaks from the first to second scan and can easily be identified. Beam damage was not observed in any apatite spectra collected here, which is consistent with previous observations that apatite is robust and does not easily incur beam damage at the energies used for S μ-XANES, even after more than 1 h of beam exposure[22,77]. However, contribution from the host zircon was observed, most severely on relatively small apatite grains (~4 μm or less in diameter). Zircon contribution to the spectra was characterized by an intense peak that begins ~2540 eV in the post-edge region, corresponding to the Zr $L\gamma_2$ transition (Supplementary Fig. 10b). The intensity of the Zr peak was less than half the intensity of the $S^{6+}$ peak in apatite in spectra collected on large apatite grains (>10 μm diameter) and the spectra was discarded if the Zr peak intensity surpassed and drowned out the $S^{6+}$ signal, illustrated in Supplementary Fig. 10b. Scans collected on host zircon showed very low sulfur counts and no characteristic edge step, therefore any contribution from zircon to the sulfur spectra is considered insignificant. Following the criteria described in ref. [22], S μ-XANES spectra were monitored for contribution of sulfur-bearing epoxy. Any spectra exhibiting contribution from other phases were discarded.

S μ-XANES spectra collected from the same apatite grain were normalized against the incident flux (I0), the pre- and post-edge set to 0 and 1, respectively, and were merged using the XAS software analysis package Athena or Ifeffit[78]. Peak positions and areas were then fitted and calculated using the Fityk software version 1.3.1[79], from which the integrated $S^{6+}/\Sigma S$ peak area ratios were calculated and input to the calibration equation in ref. [22] to calculate the values of $\log fO_2$ relative to FMQ.

**Methods in estimating melt sulfur content**. Method 1 uses experimental $D_S^{ap/m}$ calculated for a mafic melt varying with $\log fO_2$ at a temperature of ~1000 °C and pressure of ~300 MPa[22]. The effect of temperature (from ~1000 °C to the calculated model apatite saturation temperature[80]) was evaluated using the $D_S^{ap/m}$ (T) from ref. [81] that was derived for an oxidized rhyolitic melt. Although ref. [30] also reported an equation relating the $D_S^{ap/m}$ with temperature (ln ($D_S^{ap/m}$) = 21,330/T (K) −16.2), the equation does not consider the effects of depleting phosphate concentration in the melt following apatite saturation with decreasing temperature[82] and the experiment in ref. [30] conducted at ~800 °C may not represent the equilibrium S content because of early apatite crystallization from melt during annealing[81]. We therefore chose not to use it. The combined equation can be expressed as:

$$S_{ap} = D_S^{ap/m}(fO_2) \times S_{melt} \times \left[ D_S^{ap/m}(AST)/D_S^{ap/m}(1000\,°C) \right] \tag{1}$$

The values of $D_S^{ap/m}$ $(fO_2)$ are 0.85 and 1 for plutonic and volcanic rocks at FMQ + 1.24 and +1.37, whereas $D_S^{ap/m}$ $(AST)$ are estimated to be 10.2, 13.3 ± 0.2, and 10.7 at temperatures of 867 °C (HB-24, andesite porphyry), 914 ± 4 °C (HB-18 and HB-51, granodiorite porphyries; HB-32, granodiorite; and HB-30, diorite), and 951 °C (HB-33; quartz-monzonite enclave), respectively.

Method 2 to calculate the partition coefficient of S between apatite and melt is derived from ref. [82] and involves activities of phosphate, sulfate, and silica, as well as percentages of crystals, and is sensitive to the apatite saturation temperature (see details in ref. [82]). We followed the method of the linear negative relationship between $D_S^{ap/m}$ and temperature and assumed $D_S^{ap/m}$ $(AST)$ = 3.91 at 930 °C[30]. The $D_S^{ap/m}$ are estimated to be 7.7, 4.9 ± 0.2, and 2.7 at temperatures of 867 °C (HB-24, andesite porphyry), 914 ± 4 °C (HB-18 and HB-51, granodiorite porphyries; HB-32, granodiorite; and HB-30, diorite), and 951 °C (HB-33, quartz-monzonite enclave), respectively.

## Data availability
The datasets generated and discussed in this study are available in Supplementary Information (Supplementary Notes 1–4, Supplementary Figs. 1–10, Supplementary Data 1–9).

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

## Acknowledgements

The research was supported by Natural Sciences and Engineering Research Council of Canada Discovery Grant (RGPIN/5082-2017) and Canada First Research Excellence Fund through a Metal Earth thematic project to J.P.R., as well as a student research grant from the Geological Society of America and a China Scholarship Council award to X.M. We acknowledge the access permission to the drill cores and exploration database, and the logistic assistance from P. Leveille and V. Stuart-Williams at the Deep-South Resources Inc. We appreciate the assistance from D. Crabtree at Ontario GeoLabs on electron microprobe analyses. Thanks to T. Huthwelker and C. Nicoleta at SLS in Switzerland and A. Lanzirotti and M. Newville at APS in the USA for beamline assistance. We acknowledge the Paul Scherrer Institut, Villigen, Switzerland at beamline X07MB for provision of synchrotron radiation beamtime. This research also used synchrotron resources (Sector 13-ID-E) of the Advanced Photon Source, a U.S. Department of Energy (DOE) Office of Science User Facility operated for the DOE Office of Science by Argonne National Laboratory under Contract No. DE-AC02-06CH11357. S.R.T. and L.R. acknowledge financial support from NERC FAMOS NE/P01724X/1 grant and DSI-NRF Centre of Excellence, University of Johannesburg, respectively. Harquail School of Earth Sciences, Mineral Exploration Research Centre contribution MERC-ME-2021-028.

## Author contributions

X.M. and J.P.R. conceived the study and collected the samples with assistance from L.R. and G.M.B. X.M. performed petrography and sample preparation, electron microprobe analyses, LA-ICP-MS isotope and trace element analyses (with assistance from J.H.M.), as well as data compilation from literature. J.M.K. and X.M. conducted the XANES analyses. S.R.T. and R.A.S. completed zircon CA-ID-TIMS dating and O isotopes, respectively. X.M. wrote the earliest version of the manuscript as part of his Ph. D. studies with contributions from J.M.K., A.C.S., D.J.K., P.J.J. and all other co-authors.

## Competing interests

The authors declare no competing interests.
