## [Peer Review File · Nature Communications]

REVIEWER COMMENTS

Reviewer #1 (Remarks to the Author):

I found the manuscript extremely well written and convincing science. The methods applied are standard cutting edge technology and it was pleasing to see these all applied on the one ore deposit.

The major claim that will be of interest to others in the field is the very similar geochemical and geological character of the 1.88 Ga Haib PCD deposit to other PCD in the Phanerozoic, indicating that similar tectonic and ore forming processes operated at 1.88Ga as operated at 0.5 to 0 G to form PCD. To my knowledge this is the first time such a claim has been made and supported by convincing evidence.

It is very relevant to the current controversy on atmosphere oxygen levels in the Proterozoic where some scientists (eg Planavsky and co-workers) argue for very low oxygen in the Proterozoic, whereas others argue for elevated oxygen (eg Steadman et al), especially in the period 2000 to 1800 Ma, when Haib formed.

I only have one comment

There is extensive reference to Supplementary Information throughout the text. Natcomms allows more Figs. than are used in this text and i suggest that some of the important figs in supplementary Info are moved into the text, to enable easier reading by the most interested readers.

Ross Large

Reviewer #2 (Remarks to the Author):

Overall, the manuscript is very cohesive, well argued and impactful. I have few comments about the data or the interpretation, but the few I do have should be addressed before I think the manuscript can be published.

My first issue relates to the utility of using the S-valence in apatite oxybarometry method, which the authors interpret to give the most reliable fO_2 values of any oxybarometry method used in the paper. It has been demonstrated that Fe^{+3}/Fe^{+2} is easily altered at low temperatures both in nature (Helz et al. 2017) and during laboratory analyses (Cottrell et al. 2018) so I find it puzzling that the authors conclude that this is the most reliable oxybarometry method given that the samples have experienced greenschist-facies metamorphism. More justification about why you think valence could remain unaltered through all this is needed, as well as justification that XANES measurements of S give reliable results, given the problems with Fe XANES measurements (Cottrell et al. 2018).

My second issue relates to the relationship between Cu-mineralization and the studied zircons. Missing in the manuscript, aside from a brief discussion of inherited zircon, is definitive evidence that Cu-mineralization is related to crystallization of the studied zircons. Cu-mineralization is hydrothermal, and zircon crystallization is magmatic, do you expect the fO_2 of zircon crystallization to directly reflect that of exsolved hydrothermal fluids? Why or why not?

My third issue relates to novelty of the new data. It is clear that the authors have performed a diverse and large set of analyses on the samples, but what conclusions from this paper are novel? As I understand it the authors did not discover the deposit and classify it as an arc-related Cu porphyry, was it not assumed before these new data that it was formed by a similar a mechanism as seen in Phanerozoic deposits? The new fO_2 data confirm the analogous nature of the deposit to modern examples, but no time is given to previous alternative theories of formation of the deposit that the new data invalidate, and this is a disservice to the reader. The authors discuss that Cu porphyry deposits have been thought to not form in the Proterozoic, but the very existence of the studied deposit seems to contradict that statement. Are these the first data to show that the deposit is Proterozoic in age? If so, this should be made much clearer.

Once these are addressed, the manuscript should be ready for publication.

-Robert William Nicklas
Scripps Institution of Oceanography
UCSD

In Line Edits

Line 51: Rephrase this, maybe "fO₂ in log units relative to the Buffer"

Line 145: In "its" source region

Line 154: How are you this has not been reset? Valence state proxies for fO₂ are much more easily reset than elemental partitioning proxies. What evidence do you have that this has not been affected by the greenschist facies metamorphism?

Line 172: Pre-degassed is good, but when in the sequence of minerals would apatite crystallize? If it's an early formed mineral that is what you want, but you should briefly comment on the expected sequence of crystallization.

Line 181: SO₂ degassing has never been shown in a natural magma to result in oxidation, only in reduction, so this is an unsatisfying statement. What reaction in the magma would lead to oxidation with Cl degassing? Do you expect degassing to be relevant for a plutonic rock as you are studying here? I suspect the different proxies would be closer in the valence state proxy were more reliable.

Line 454: "Ground" not "grinded"

Line 625: Did you do XANES measurements on apatites that had been previously analyzed on the microprobe? Could there be an effect of the electron beam on the measure S valence? Long count times under an electron beam have been found to alter Fe valences in glasses (Zhang et al. 2018).

Line 653: How do you account for beam damage during your XANES measurements? This has been found to be significant for Fe valences (Cottrell et al. 2018), especially at the very small scale you are working with.

Works Cited

- Cottrell E., Lanzirotti A., Mysen B., Birner S., Kelley K. A., Botcharnikov R., Davis F. A., Newville M. (2018) A Mössbauer-based XANES calibration for hydrous basalt glasses reveals radiation-induced oxidation of Fe. *American Mineralogist* 103: 489-501.
- Helz, R. T., Cottrell, E., Brounce, M. N., Kelley, K. A. (2017) Olivine-melt relationships and syneruptive redox variations in the 1959 eruption of Kilauea Volcano as revealed by XANES. *Journal of Volcanology and Geothermal Research* 333-334: 1-14.
- Zhang, C., Almeev, R. R., Hughs, E. C., Borisov, A. A., Wolff, E. P., Hofer, H. E., Botcharnikov, R. E., Koepke, J. (2018). Electron microprobe technique for the determination of iron oxidation state in silicate glasses. *American Mineralogist* 103: 1445-1454.

Reviewer #3 (Remarks to the Author):

I have read the revised manuscript, "Oxidized, sulfur-rich arc magmas formed porphyry Cu deposits by 1.88 Ga" by Xuyang Meng and co-authors.

Meng and co-authors examine whole rock major element compositions, zircon major and trace elements and Hf and O isotopes, and apatite major element and S oxidation states of samples from the Haib Cu porphyry in the Paleoproterozoic Richertsveld Magmatic Arc. These data are used to assess the importance of elevated fO₂, S, and various differentiation processes during the formation of the Cu porphyry as a direct test of several recent hypotheses surrounding the origin of oxidized arc magmas and Cu porphyry deposits.

I am supportive of the work, but I think it would be better suited to a longer format journal, where the good and careful work of Meng et al. can be fully described in the main text, and the speculative arguments at the end of the manuscript can be discussed more fully.

1. Minor comments:

Line 66 – "...and potentially limiting the fertility of..." -> "...and potentially limiting the ore forming potential of..."

Line 68: "The aforementioned hypothesis..." -> unclear what this refers to.

Line 75: what is "weak" metamorphism? Low-grade? Localized? Some combination of these?

Line 76: I do not know the term "causative magmas". It may be that the broad readership of this journal also does not.

Line 80: "Contrary to previous knowledge"...what previous knowledge...of the Precambrian? Provide a citation?

2. A case for a longer format of this manuscript

Beginning at line 142 with the oxygen isotope discussion, I found that the main text was too brief in describing the excellent dataset presented in this work. I understand that this is because of strict character counts required by the journal, but I do not think the minimum amount of description of the data is provided. The oxygen isotope story should be described to the reader – the values are overlapping with the values expected for DMM and extend to heavier values, indicating some level of communication with near surface-derived materials. How much assimilation is necessary of material with what oxygen isotope composition to explain the heavy values obtained? The main text points to Fig S2, but these are sample photographs and the supplemental text otherwise does not describe any such calculations.

The oxidation state of sulfur measurements in the apatite inclusions within zircon are strong evidence for an oxidized system at this time, which is the interesting observation presented. However, there is an important detail about the relative timing of apatite and zircon crystallization and volatile saturation that the authors describe around lines 180, but the arguments require the supplementary text to understand and are generally weak because of the limited nature of their constraints. A longer manuscript could describe this adequately, including why they are discussing the oxidizing effects of Cl degassing (the observation of Metrich re: the oxidizing effects of S degassing have not been born out – see various works from Moussallam, Brounce, Shorttle, Hartley, Helz on OIB, and the Bell work on Cl degassing is not widely known – the authors must describe in at least one sentence what Aaron's idea is and why it might explain the dataset on hand). Again, the primary observation from zircon oxybarometry is that the system is oxidized at the time of apatite and zircon crystallization.

The sulfur abundance calculations again require reading the supplement to understand even the basic approach taken here – some of the supplementary text must be in the main text so that the reader can get the basic idea. Details can be put in the supplement, but not the full description of what is being done.

I also disagree with the characterization that "Phanerozoic arc melts" have high sulfur (~line 199) – not only are there a few instances where arc-related basalts and boninites have low sulfur and maybe as the result of a mantle source depleted in sulfur (see Brounce et al., G-cubed 2017; there are a handful of other observations like this from Cooper, Rasmussen, Vastelich), but because the statement is worded so broadly, it is wrong about any melt that has partially or fully degassed with respect to sulfur, which is every melt that ascends to low enough pressure in volcanic edifices. I am fairly certain that the authors do not mean to include partially or fully degassed melts into this characterization of Phanerozoic arc magmas, but what is written is so broad as to be wrong in these cases.

Being an apatite and Cu-porphyry paper, and given the recent interest in this topic, I think it is critically important to develop fully the argument against significant garnet fractionation described around line 215. The statement at line 220 "Local assimilation...could not have changed the magmatic fO_2 ..." I do not understand, and should be removed or backed by quantitative treatment of the relative effects of major composition and Fe, S, etc redox states on magmatic fO_2 .

Everything from line 226 is extremely speculative, and a little circular. The authors state, "This makes sense, because the oceans were full of sulfate at this time." but then use the oxidized nature of these rocks to mildly suggest that this may have "influenced the atmosphere-oceanic redox state". Which is first, the chicken or the egg?

I think the data are excellent, the interpretations may even be correct, but the arguments in support of these interpretations need to be developed more carefully in the main text. It may be that this is significantly easier for the authors in a longer format manuscript.

I hope that you find my comments here helpful. Best wishes for all, I hope everyone is well.

-Maryjo Brounce

RESPONSE TO REVIEWERS

Oxidized, sulfur-rich arc magmas formed porphyry Cu deposits by 1.88 Ga

We would like to thank the reviewers Ross Large, Robert Nicklas, and Maryjo Brounce for their constructive reviews. By addressing their concerns and accommodating their suggestions, we consider the revised version to be more rigorous and the ideas easier to follow and understand. In addition, we have reformatted the manuscript following the requirement for Nature Communications and made a few edits which can be seen in the submitted manuscript with changes tracked.

Response to Reviewer #1 's comments:

We appreciate Dr. Ross Large's recognition of the scientific significance of this research. Following the comments, we now include a few supplementary figures in the previously submitted version in the main text. See details in the reply as follows.

"I found the manuscript extremely well written and convincing science. The methods applied are standard cutting edge technology and it was pleasing to see these all applied on the one ore deposit.

The major claim that will be of interest to others in the field is the very similar geochemical and geological character of the 1.88 Ga Haib PCD deposit to other PCD in the Phanerozoic, indicating that similar tectonic and ore forming processes operated at 1.88Ga as operated at 0.5 to 0 G to form PCD. To my knowledge this is the first time such a claim has been made and supported by convincing evidence.

It is very relevant to the current controversy on atmosphere oxygen levels in the Proterozoic where some scientists (eg Planavsky and co-workers) argue for very low oxygen in the Proterozoic, whereas others argue for elevated oxygen (eg Steadman et al), especially in the period 2000 to 1800 Ma, when Haib formed.

I only have one comment

There is extensive reference to Supplementary Information throughout the text. Natcomms allows more Figs. than are used in this text and I suggest that some of the important figs in supplementary Info are moved into the text, to enable easier reading by the most interested readers."

We agree with the reviewer that some important references and figures in the Supplementary Information can be moved to the text. (1) We moved the simplified

geological map of the Haib area (originally Fig. S1a, now Fig. 1) to the main text along with the rare-earth element spider diagram and its inset (originally Fig. S3d, now Fig. 2) from the Supplementary Information. The geological map helps readers better understand the geological background and sample locations. The rare-earth element spider diagram and the inset diagram of Dy/Yb versus SiO₂ is discussed in the 'Discussion and Conclusion' section to suggest amphibole- versus garnet-dominated fractionation. Fig. 2 in the revised manuscript is essential because garnet fractionation has recently been hypothesized to make arc magmas relatively oxidized (Tang et al., 2018; Lee et al., 2020). We use the diagrams to show lack of garnet fractionation for the Haib magmas. (2) The method for estimating melt S concentration from apatite S contents has also been moved to the main text, hence the number of the references can significantly be reduced in the Supplementary Information.

References:

Tang M, Erdman M, Eldridge G, Lee CTA. The redox "filter" beneath magmatic orogens and the formation of continental crust. *Science Advances* **4**, eaar4444 (2018).
Lee CTA, Tang M. How to make porphyry copper deposits. *Earth and Planetary Science Letters* **529**, 115868 (2020).

Response to Reviewer #2's comments:

We thank Dr. Robert W. Nicklas's concerns about the methods and scientific significance. Our replies are itemized following each comment.

"Overall, the manuscript is very cohesive, well argued and impactful. I have few comments about the data or the interpretation, but the few I do have should be addressed before I think the manuscript can be published.

My first issue relates to the utility of using the S-valence in apatite oxybarometry method, which the authors interpret to give the most reliable fO₂ values of any oxybarometry method used in the paper. It has been demonstrated that Fe⁺³/Fe⁺² is easily altered at low temperatures both in nature (Helz et al. 2017) and during laboratory analyses (Cottrell et al. 2018) so I find it puzzling that the authors conclude that this is the most reliable oxybarometry method given that the samples have experienced greenschist-facies metamorphism. More justification about why you think valence could remain unaltered through all this is needed, as well as

justification that XANES measurements of S give reliable results, given the problems with Fe XANES measurements (Cottrell et al. 2018).”

(1) We agree with the reviewer that irradiation-induced beam damage of silicate glass challenges accurate determination of melt $\text{Fe}^{3+}/\text{Fe}^{2+}$ ratios and related $f\text{O}_2$ calculation. However, the beam energy used for S-XANES analyses is an order of magnitude lower than the energy required for Fe-XANES, and the S oxidation states in apatite (mineral phase versus melt) have been shown not to be affected by extended duration of exposure to a high-flux photon beam (Konecke et al., 2017).

Because beam damage manifests as the growth or shrinkage of absorption peaks from the first to second scan and can readily be detected, we collected two scans for each point to monitor beam damage by irradiation of the focused X-ray beam for our samples. The spectra we collected (for samples HB-18, HB-24, HB-28, HB-30, HB-32, and HB-51) do not show evidence of beam damage between the first and second scan, an observation that is consistent with the findings of Konecke et al. (2017) and is evidence that the results are not an analytical artefact. This information is now described in the ‘Methods’ section.

(2) Apatite can accommodate different sulfur species through the following mechanisms: $2\text{P}^{5+} = \text{S}^{6+} + \text{S}^{4+}$ and $2(\text{F}, \text{Cl}, \text{OH})^- = \text{S}^{2-} + \square$ (Kim et al., 2017). Because the mineral apatite has a well-defined crystallographic structure, the sulfur species and their proportion will remain constant. Considering the apatite grains that we analyzed are inclusions protected by zircon (a mineral phase that is highly resistant to metamorphism and alteration), the primary composition of apatite can therefore be preserved (Jennings et al., 2011; Bell et al., 2016). The measured U-Pb isotopic and trace element compositions of the primary zircon hosts (with oscillatory and sector zonings; see representative CL images in Supplementary Figure 5) from the Haib rocks are uniform and consistent with a closed system after their crystallization. The evidence further supports the conclusion that greenschist metamorphism and hydrothermal fluids did not affect the zircon grains and that the apatite inclusions preserve a primary chemistry.

Based on these lines of evidence, we argue that the S-in-apatite oxybarometer used for the zircon-hosted apatite inclusions provides a reliable $f\text{O}_2$ estimation for our study.

References:

Konecke BA, Fiege A, Simon AC, Parat F, Stechern A. Co-variability of S^{6+} , S^{4+} , and S^{2-} in apatite as a function of oxidation state: Implications for a new oxybarometer. *American Mineralogist* **102**, 548–557 (2017).

Kim Y, Konecke B, Fiege A, Simon A, Becker U. An ab-initio study of the energetics and geometry of sulfide, sulfite, and sulfate incorporation into apatite: The thermodynamic basis for using this system as an oxybarometer. *American Mineralogist* **102**, 1646–1656 (2017).

Jennings ES, Marschall H, Hawkesworth C, Storey C. Characterization of magma from inclusions in zircon: Apatite and biotite work well, feldspar less so. *Geology* **39**, 863-866 (2011).

Bell EA. Preservation of primary mineral inclusions and secondary mineralization in igneous zircon: a case study in orthogneiss from the Blue Ridge, Virginia. *Contributions to Mineralogy and Petrology* **171**, 26 (2016).

“My second issue relates to the relationship between Cu-mineralization and the studied zircons. Missing in the manuscript, aside from a brief discussion of inherited zircon, is definitive evidence that Cu-mineralization is related to crystallization of the studied zircons. Cu-mineralization is hydrothermal, and zircon crystallization is magmatic, do you expect the fO_2 of zircon crystallization to directly reflect that of exsolved hydrothermal fluids? Why or why not?”

This is a very valid comment and we welcome the chance to respond to this important point. As suggested, we do not expect fO_2 at the time of zircon crystallization to directly reflect that of the later exsolved hydrothermal fluids. Instead, we focus on magmatic fO_2 as this is a more critical factor that affects the ore-forming potential of the magmas.

Porphyry Cu deposits are considered to form from magmatic-hydrothermal fluids exsolved from silicate melt in a subjacent magma chamber (Sillitoe et al., 2010). Depletion of Cu in the silicate melt will reduce its ore-forming potential, and such cases readily occur in relatively reduced melts because of voluminous early sulfide saturation during magma evolution that scavenges Cu from the melt (Simon and Ripley, 2011). In contrast, in a relatively oxidized environment, Cu behaves as an incompatible element during magma evolution and thus it can be transported in the melt to the causative magma chamber and partition into the fluid phase when the hydrothermal fluid exsolves. Our high-precision zircon U-Pb geochronological results demonstrate that the Haib porphyry Cu mineralization coincides with the formation of the Vioorsdrif Intrusive Complex, which strongly suggests it was the causative magma that provided Cu for the Haib porphyry-type mineralization. It is for this reason that we studied the nature of the syn-mineralization Vioorsdrif Intrusive Complex.

References:

Sillitoe RH. Porphyry Copper Systems. *Economic Geology* **105**, 3–41 (2010).

Simon, A.C. and Ripley, E. The role of magmatic sulfur in the formation of ore deposits. In *Sulfur in Magmas and Melts: Its Importance for Natural and Technical Processes* (eds Behrens, H. & Webster, J. D.) *Reviews in Mineralogy and Geochemistry*, Mineralogical Society of America **73**, 513–578 (2011).

My third issue relates to novelty of the new data. It is clear that the authors have performed a diverse and large set of analyses on the samples, but what conclusions from this paper are novel? As I understand it the authors did not discover the deposit and classify it as an arc-related Cu porphyry, was it not assumed before these new data that it was formed by a similar a mechanism as seen in Phanerozoic deposits? The new fO₂ data confirm the analogous nature of the deposit to modern examples, but no time is given to previous alternative theories of formation of the deposit that the new data invalidate, and this is a disservice to the reader. The authors discuss that Cu porphyry deposits have been thought to not form in the Proterozoic, but the very existence of the studied deposit seems to contradict that statement. Are these the first data to show that the deposit is Proterozoic in age? If so, this should be made much clearer.

Once these are addressed, the manuscript should be ready for publication.”

The reviewer has raised several valid points. We will address these points as follows.

“It is clear that the authors have performed a diverse and large set of analyses on the samples, but what conclusions from this paper are novel?”

In this manuscript, we claim that oxidized, sulfur-rich arc magmas formed porphyry Cu deposits in the Paleoproterozoic based on our robust results. We highlight the novelty in our title “Oxidized sulfur-rich arc magmas formed porphyry Cu deposits by 1.88 Ga”. We will explain the novelty by addressing a few other issues the reviewer raised.

“As I understand it the authors did not discover the deposit and classify it as an arc-related Cu porphyry, was it not assumed before these new data that it was formed by a similar a mechanism as seen in Phanerozoic deposits?” and the last question *“Are these the first data to show that the deposit is Proterozoic in age? If so, this should be made much clearer.”*

It is correct that we did not discover the Haib deposit or lay claim to being the first to suggest its analogue to Phanerozoic porphyry-type systems, but it is our new high-quality U-Pb geochronology and litho-geochemistry data that (1) robustly constrains

the temporal and genetic relationship between porphyry Cu mineralization and the porphyritic and equigranular plutonic rocks and (2) reinforces the classification of the Haib deposit as an arc-related Cu porphyry. A detailed deposit description is provided in the cited references in the main text and is not the focus of this study. What we addressed is a broader and more relevant scientific issue: whether similar metallogenic processes for the Phanerozoic porphyry Cu deposits also operated in the Precambrian when Haib formed. We agree with the reviewer that it is *assumed* that a similar mechanism for forming a porphyry Cu deposit operated through geological time, but this assumption exists because it remains unconstrained.

“The new fO_2 data confirm the analogous nature of the deposit to modern examples, but no time is given to previous alternative theories of formation of the deposit that the new data invalidate, and this is a disservice to the reader. The authors discuss that Cu porphyry deposits have been thought to not form in the Proterozoic, but the very existence of the studied deposit seems to contradict that statement.”

The current understanding of the metallogenic processes for porphyry Cu deposits is developed based on many studies on Phanerozoic deposits, but it remains unclear as to whether similar metallogenic processes also operated in the Precambrian. Several members of the scientific community suggest that the tectonomagmatic conditions of Precambrian arc settings related to oceanic-atmospheric conditions do not support formation of such deposits in the Precambrian (Richards and Mumin, 2013; Evans and Tomkins, 2011). This knowledge gap is difficult to fill for three reasons: (1) the genesis of a few so-called ‘porphyry Cu deposits’ remain disputed because overprinting high-grade metamorphism and extensive deformation modified the primary mineralogy and textures, thus making identification of the genesis difficult (please see the criteria for identifying porphyry Cu deposit in Sillitoe et al., 2010); (2) tectonic disturbances and resulting resetting of isotopic systematics make identifying the causative magmas difficult (Meng et al., 2020); and (3) the modification of primary mineralogy due to metamorphism and alteration, combined with the lack of suitable oxybarometers make constraining magmatic fO_2 difficult, which relates to this aspect noted above. The information regarding the difficulty in filling the knowledge gap is summarized in the ‘Introduction’ section.

In this study, we targeted the Haib deposit to circumvent the above problems because the deposit experienced little deformation and relatively low-grade metamorphism which greatly enhances identifying the parameters and criteria needed for identifying a porphyry Cu deposit (see the third paragraph of the ‘Introduction’ section). We used high-precision geochronological methods to date the pre-ore, syn-ore, and post-ore igneous rocks and hence bracket the timing of the mineralization (see the ‘High-precision U-Pb geochronology’ part of the ‘Results’ section in the revised manuscript). This allowed us to convincingly show for the first

time that the plutonic rocks in the Vioorsdrif Complex are the source for the mineralization. Furthermore, we used recently calibrated, novel oxybarometers to constrain fO_2 , as well as melt S contents for the magmas (see the 'Results' section in the revised manuscript). The latter data therefore provide the basis to identify and argue that similar metallogenic processes for the Phanerozoic porphyry Cu deposits did indeed operate in the Precambrian. Furthermore, this supports the idea of oxidized seafloor sediments in the oceanic lithosphere in the Paleoproterozoic (see the 'Discussion and conclusion' section) and contributes to the ongoing debate of whether the surface environment in the Paleoproterozoic was oxidized (e.g., Lyons et al., 2014; Steadman et al., 2020).

As such, we hope the reviewer recognizes that our results are novel and the data are technically sound using state-of-art technologies (CA-ID-TIMS, SIMS, and μ -XANES).

References:

Sillitoe RH. Porphyry Copper Systems. *Economic Geology* **105**, 3–41 (2010).

Meng XY, *et al.* The Tongkuangyu Cu Deposit, Trans-North China Orogen: A Metamorphosed Paleoproterozoic Porphyry Cu Deposit. *Economic Geology* **115**, 51–77 (2020).

Lyons TW, Reinhard CT, Planavsky NJ. The rise of oxygen in Earth's early ocean and atmosphere. *Nature* **506**, 307–315 (2014).

Steadman JA, *et al.* Evidence for elevated and variable atmospheric oxygen in the Precambrian. *Precambrian Research*, 105722 (2020).

In Line Edits

Line 51: Rephrase this, maybe " fO_2 in log units relative to the Buffer"

Resolved.

Line 145: In "its" source region

Resolved.

Line 154: How are you this has not been reset? Valence state proxies for fO_2 are much more easily reset than elemental partitioning proxies. What evidence do you have that this has not been affected by the greenschist facies metamorphism?

Please see the reply to the comment #1.

Line 172: Pre-degassed is good, but when in the sequence of minerals would apatite crystallize? If it's an early formed mineral that is what you want, but you should briefly comment on the expected sequence of crystallization.

We have briefly interpreted the crystallization sequence in Line 156–158 (original version), where the apatite grains are interpreted to crystallize under near-liquidus conditions as early crystallized mineral phases. Please see the Line 168–171 in the new version.

Line 181: SO₂ degassing has never been shown in a natural magma to result in oxidation, only in reduction, so this is an unsatisfying statement. What reaction in the magma would lead to oxidation with Cl degassing? Do you expect degassing to be relevant for a plutonic rock as you are studying here? I suspect the different proxies would be closer in the valence state proxy were more reliable.

In the revised version, such interpretation is deleted considering testing the effects of degassing of SO₂ and Cl on redox state of the magmas in Haib is not the focus of this study. We instead focus on the consistent, relatively high fO_2 values of the magmatic system. We agree with the reviewer that SO₂ degassing results in reduction versus oxidation of most reported natural magmas, but in theory, the degassing can lead to either oxidation or reduction of the residual melt depending on speciation of S in the melt and gas. Degassing of Cl can oxidize the residual melt by leaching Fe²⁺ in Cl-rich hydrothermal fluids (Bell and Simon, 2011).

Reference:

Bell, A. and Simon, A.C. (2011) Evidence for the alteration of the Fe³⁺/ΣFe of silicate melt caused by the degassing of chlorine-bearing aqueous volatiles. *Geology*. 39(5), 499-502.

Line 454: “Ground” not “grinded”

Resolved. Thanks for catching this.

Line 625: Did you do XANES measurements on apatites that had been previously analyzed on the microprobe? Could there be an effect of the electron beam on the measure S valence? Long count times under an electron beam have been found to alter Fe valences in glasses (Zhang et al. 2018).

We conducted μ -XANES analyses on a different group of apatite inclusions, which were not used for electron microprobe analyses, in order to eliminate beam damage from electron microprobe. We add the details in the method of *Micro X-ray absorption near-edge structure spectroscopy (μ -XANES)*.

Line 653: How do you account for beam damage during your XANES measurements? This has been found to be significant for Fe valences (Cottrell et al. 2018), especially at the very small scale you are working with.

We scanned each point twice to monitor beam damage, and saw no changes in the spectra from the first to second scan. Beam damage from μ -XANES is not as big of a concern when analyzing solid mineral phases compared to measuring silicate melt, and the energies used for S-XANES are an order of magnitude lower than those used for Fe-XANES. Please see the details in the reply to comment #1.

Works Cited

- Cottrell E., Lanzirotti A., Mysen B., Birner S., Kelley K. A., Botcharnikov R., Davis F. A., Newville M. (2018) A Mössbauer-based XANES calibration for hydrous basalt glasses reveals radiation-induced oxidation of Fe. *American Mineralogist* 103: 489-501.
- Helz, R. T., Cottrell, E., Brounce, M. N., Kelley, K. A. (2017) Olivine-melt relationships and syneruptive redox variations in the 1959 eruption of Kīlauea Volcano as revealed by XANES. *Journal of Volcanology and Geothermal Research* 333-334: 1-14.
- Zhang, C., Almeev, R. R., Hughs, E. C., Borisov, A. A., Wolff, E. P., Hofer, H. E., Botcharnikov, R. E., Koepke, J. (2018). Electron microprobe technique for the determination of iron oxidation state in silicate glasses. *American Mineralogist* 103: 1445-1454.

Response to Reviewer #3's comments:

Many thanks to Dr. Maryjo Brounce for comments on some details and the suggestion of extending the length of the manuscript. We have extended the parts following most of the comments. Our replies are itemized following each comment.

"I have read the revised manuscript, "Oxidized, sulfur-rich arc magmas formed porphyry Cu deposits by 1.88 Ga" by Xuyang Meng and co-authors.

Meng and co-authors examine whole rock major element compositions, zircon major and trace elements and Hf and O isotopes, and apatite major element and S

oxidation states of samples from the Haib Cu porphyry in the Paleoproterozoic Richtersveld Magmatic Arc. These data are used to assess the importance of elevated fO₂, S, and various differentiation processes during the formation of the Cu porphyry as a direct test of several recent hypotheses surrounding the origin of oxidized arc magmas and Cu porphyry deposits.

I am supportive of the work, but I think it would be better suited to a longer format journal, where the good and careful work of Meng et al. can be fully described in the main text, and the speculative arguments at the end of the manuscript can be discussed more fully.”

We appreciate the reviewer for recognizing the significance of our research. As reminded by the editor, Nature Communications allows a longer format for the manuscript and thus enables us to resolve the reviewer’s comments. We appreciate the reviewer’s comments and have extended those parts where we agree with the reviewer as to their importance.

1. Minor comments:

Line 66 – “...and potentially limiting the fertility of...” -> “...and potentially limiting the ore forming potential of...”

Resolved. We changed “...and potentially limiting the fertility of...” to “...and limiting the ore-forming potential of...”.

Line 68: “The aforementioned hypothesis...” -> unclear what this refers to.

We have added ‘that tectonomagmatic conditions in the Precambrian are unfavorable for porphyry Cu deposit formation’ following ‘The hypothesis’.

Line 75: what is “weak” metamorphism? Low-grade? Localized? Some combination of these?

We meant ‘Low-grade’ metamorphism. Have changed in the manuscript accordingly.

Line 76: I do not know the term “causative magmas”. It may be that the broad readership of this journal also does not.

We changed ‘causative magmas for mineralization’ to ‘ore-related magmas’.

Line 80: “Contrary to previous knowledge”...what previous knowledge...of the Precambrian? Provide a citation?

We added references of Richards and Mumin, 2013 and Evans and Tomkin, 2011 for ‘Contrary to previous knowledge’.

2. A case for a longer format of this manuscript

“Beginning at line 142 with the oxygen isotope discussion, I found that the main text was too brief in describing the excellent dataset presented in this work. I understand that this is because of strict character counts required by the journal, but I do not think the minimum amount of description of the data is provided. The oxygen isotope story should be described to the reader – the values are overlapping with the values expected for DMM and extend to heavier values, indicating some level of communication with near surface-derived materials. How much assimilation is necessary of material with what oxygen isotope composition to explain the heavy values obtained? The main text points to Fig S2, but these are sample photographs and the supplemental text otherwise does not describe any such calculations.”

a. Beginning at line 142 with the oxygen isotope discussion, I found that the main text was too brief in describing the excellent dataset presented in this work. I understand that this is because of strict character counts required by the journal, but I do not think the minimum amount of description of the data is provided. The oxygen isotope story should be described to the reader – the values are overlapping with the values expected for DMM and extend to heavier values, indicating some level of communication with near surface-derived materials.

Thanks for the suggestion and the zircon oxygen isotope story was expanded in the main text. Because the main purpose of zircon Hf-O isotopes was to constrain the homogeneity of the magmatic source and to preclude crustal assimilation during magma ascent which may affect the redox state of the magmas, to make the story more complete, we added:

“The elevated zircon $\delta^{18}\text{O}$ values for the volcanic and porphyritic rocks compared to the equigranular rocks indicate more contamination from the recycled upper crustal materials that had undergone low-temperature alteration (Valley *et al.*, 2005). The near-constant Hf isotope ratios among the samples (Fig. 3b) suggest a limited effect of the crustal materials on the Hf isotopic systematics of the source and also support insignificant crustal assimilation during magma ascent.”

Reference:

Valley JW, *et al.* 4.4 billion years of crustal maturation: oxygen isotope ratios of magmatic zircon. *Contributions to Mineralogy and Petrology* **150**, 561–580 (2005).

b. How much assimilation is necessary of material with what oxygen isotope composition to explain the heavy values obtained?

We would be glad to address this question. However, modelling the degree of sedimentary contamination requires accurate $\delta^{18}\text{O}$ data for the sedimentary rocks, lacking $\delta^{18}\text{O}$ data for pre-Haib sedimentary rocks nearby RMA (Richtersveld Magmatic Arc) makes constraining the degrees of the sedimentary contamination difficult. If we assume the recycled near-surface crustal materials yielded similar $\delta^{18}\text{O}$ values to that for global Paleoproterozoic fine-grained siliciclastic sediments (~14 ‰ on average; Bindeman *et al.*, 2016), a modeled ~16 % and ~6 % sedimentary contamination of the source region for the equigranular rocks (using their average zircon O isotope value as a reference) would be required to obtain the zircon $\delta^{18}\text{O}$ values for the volcanic and porphyritic rocks, respectively.

We aim to make the science as robust as possible and therefore remain cautious about reporting an estimation based mainly on an assumption. We decided not to put the above argument in the manuscript, particularly because lacking the estimation will not affect our conclusion.

Reference:

Bindeman I, Bekker A, Zakharov D. Oxygen isotope perspective on crustal evolution on early Earth: A record of Precambrian shales with emphasis on Paleoproterozoic glaciations and Great Oxygenation Event. *Earth and Planetary Science Letters* **437**, 101–113 (2016).

c. The main text points to Fig S2, but these are sample photographs and the supplemental text otherwise does not describe any such calculations.

The purpose of Figure S2 is to provide petrographic evidence suggesting assimilation of volcanic rocks to the porphyritic rocks. We deleted the argument because the minor amounts of assimilation of the volcanic rocks is not sufficient to change the $\delta^{18}\text{O}$ values of the porphyritic rocks. Please see the revised version.

References:

Valley JW, *et al.* 4.4 billion years of crustal maturation: oxygen isotope ratios of magmatic zircon. *Contributions to Mineralogy and Petrology* **150**, 561–580 (2005).

Bindeman I, Bekker A, Zakharov D. Oxygen isotope perspective on crustal evolution on early Earth: A record of Precambrian shales with emphasis on Paleoproterozoic glaciations and Great Oxygenation Event. *Earth and Planetary Science Letters* **437**, 101–113 (2016).

“The oxidation state of sulfur measurements in the apatite inclusions within zircon are strong evidence for an oxidized system at this time, which is the interesting observation presented. However, there is an important detail about the relative timing of apatite and zircon crystallization and volatile saturation that the authors describe around lines 180, but the arguments require the supplementary text to understand and are generally weak because of the limited nature of their constraints. A longer manuscript could describe this adequately, including why they are discussing the oxidizing effects of Cl degassing (the observation of Metrich re: the oxidizing effects of S degassing have not been born out – see various works from Moussallam, Brounce, Shorttle, Hartley, Helz on OIB, and the Bell work on Cl degassing is not widely known – the authors must describe in at least one sentence what Aaron’s idea is and why it might explain the dataset on hand). Again, the primary observation from zircon oxybarometry is that the system is oxidized at the time of apatite and zircon crystallization.”

Thanks for the suggestions. We have added the basic approach of identifying the relative timing of apatite and volatile saturation in the main text and leave some details in the Supplementary Note 2 (for the revised manuscript). With the available dataset, the analyzed apatite grains can be interpreted to have crystallized in a volatile-undersaturated environment. Also, we deleted the sentences regarding the potentially oxidizing effects of SO₂ and Cl degassing, because they may be contentious and largely irrelevant to the main points of the paper. Thus, we do not include an explanation of the idea of Bell and Simon *et al.* (2011) regarding the effect of Cl degassing on the magmatic redox state in the main text (that is, “exsolution of saline fluids, in which Fe²⁺ is more soluble than Fe³⁺, results in increasing Fe³⁺/ΣFe ratios of the residual magmas”). We instead focus on the consistent, relatively high *f*O₂ results from the different methods, as suggested by the reviewer.

Reference:

Bell AS, Simon A. Experimental evidence for the alteration of the Fe³⁺/ΣFe of silicate melt caused by the degassing of chlorine-bearing aqueous volatiles. *Geology* **39**, 499–502 (2011).

“The sulfur abundance calculations again require reading the supplement to understand even the basic approach taken here – some of the supplementary text must be in the main text so that the reader can get the basic idea. Details can be put in the supplement, but not the full description of what is being done.”

We agree with the reviewer on this comment. We have put the basic approach in the main text and the details in the ‘Methods’ section.

“I also disagree with the characterization that “Phanerozoic arc melts” have high sulfur (~line 199) – not only are there a few instances where arc-related basalts and boninites have low sulfur and maybe as the result of a mantle source depleted in sulfur (see Brounce et al., G-cubed 2017; there are a handful of other observations like this from Cooper, Rasmussen, Vastelich), but because the statement is worded so broadly, it is wrong about any melt that has partially or fully degassed with respect to sulfur, which is every melt that ascends to low enough pressure in volcanic edifices. I am fairly certain that the authors do not mean to include partially or fully degassed melts into this characterization of Phanerozoic arc magmas, but what is written is so broad as to be wrong in these cases.”

We agree with the reviewer that not all Phanerozoic arc melts are S-rich and some S-poor arc-related basalts exist. This argument has also been supported by the histogram in Fig. 5b (the revised manuscript; replacing Fig. 3b in the original version), in which some Phanerozoic arc melt inclusions have relatively low S contents. We thank the reviewer for pointing out this broad statement in the main text. Because most data show relatively high S contents for the Phanerozoic arc melts, we choose to add ‘many’ before ‘Phanerozoic arc melts’ in the main text. In addition, because the compiled data are for melt inclusions interpreted to have experienced limited degassing, we add the constraint in the parentheses. In Fig. 5b in the revised manuscript, we only include melt inclusion data with $\text{CO}_2 \geq 50$ ppm for arc rocks that are suggested to have insignificantly affected by degassing. Replotting the data reveals a broadly similar distribution of melt S contents with that in the original version. The average S value has been changed to 0.09 ± 0.07 wt. % (1σ). The related information has also been added in the caption of the Fig. 5 in the revised manuscript.

“Being an apatite and Cu-porphyry paper, and given the recent interest in this topic, I think it is critically important to develop fully the argument against significant garnet fractionation described around line 215. The statement at line 220 “Local assimilation...could not have changed the magmatic $f\text{O}_2$...” I do not understand, and

should be removed or backed by quantitative treatment of the relative effects of major composition and Fe, S, etc redox states on magmatic fO_2 .”

We agree with the reviewer on this comment, and we have developed the argument against significant garnet fractionation in the ‘Discussion and Conclusion’ section. We added “Deep crustal garnet fractionation has recently been hypothesized to auto-oxidize magma for porphyry Cu deposit formation (Tang et al., 2018; Lee et al., 2020). However, the Haib magmas yielded listric REE patterns and moderately negative Eu anomalies which indicate low-pressure differentiation with amphibole \pm plagioclase fractionation (Fig. 2). Amphibole preferentially incorporates middle REEs over heavy REEs and garnet incorporates heavy REEs relative to light REEs, thus amphibole fractionation will decrease, but garnet fractionation will increase, residue melts of Dy/Yb ratios (Macpherson et al., 2006; Davidson et al., 2007). The decreasing Dy/Yb ratios with increasing SiO₂ content (Fig. 2) support amphibole-dominated fractionation without or with minimal garnet influence, leading us to discount the possibility that the elevated magmatic fO_2 of the Haib magmas is a result of garnet fractionation.”

We also deleted the sentence of “Local assimilation...could not have changed the magmatic fO_2 ...” following the suggestion.

References:

Tang M, Erdman M, Eldridge G, Lee CTA. The redox “filter” beneath magmatic orogens and the formation of continental crust. *Science Advances* **4**, eaar4444 (2018).

Lee CTA, Tang M. How to make porphyry copper deposits. *Earth and Planetary Science Letters* **529**, 115868 (2020).

Macpherson CG, Dreher ST, Thirlwall MF. Adakites without slab melting: High pressure differentiation of island arc magma, Mindanao, the Philippines. *Earth and Planetary Science Letters* **243**, 581–593 (2006).

Davidson J, Turner S, Handley H, Macpherson C, Dosseto A. Amphibole “sponge” in arc crust? *Geology* **35**, 787–790 (2007).

“Everything from line 226 is extremely speculative, and a little circular. The authors state, “This makes sense, because the oceans were full of sulfate at this time.” but then use the oxidized nature of these rocks to mildly suggest that this may have “influenced the atmosphere-oceanic redox state”. Which is first, the chicken or the egg?”

We appreciate the reviewer for pointing out this. While the last sentence (‘If such oxidized and sulfur-rich magmas were common on Earth at 1.88 Ga, the degassed

volatiles may have greatly influenced the atmosphere-oceanic redox states.”) can be a bit speculative because (a) we assumed that such oxidized and sulfur-rich magmas are common on Earth at 1.88 Ga, and (b) the effect of volcanic degassing on atmospheric redox state is complex (Holland et al., 2002; Brounce et al., 2017), we do not think however that everything from line 226 (original version) is extremely speculative. From Line 226 of the main text (original version), we are exploring the origin of the oxidized fluids dehydrated from the slab in the context of current understanding of related topics, and each statement is well supported by literature. Considering the oceanic basalts are relatively reduced (poor in Fe³⁺) that may not contribute sufficient oxidized materials to the mantle (Stolper et al., 2018), we argue that the most plausible agent is sulfate, a more efficient oxidant as demonstrated by Kelley and Cottrell (2009).

- (1) We apologize for implying to the reviewer that “the oceans were full of sulfate at this time” from our statement. What we wanted to highlight and indicate is that submarine sulfate (no more than the average sulfur contents in the modern ocean) can be recycled to oxidize the sub-arc mantle based on evidence provided by recent studies on measuring marine sulfate content in the Paleoproterozoic.
- (2) Resolving ‘Chicken-egg’ doubt is out of the scope of this study. How the pre-1.88 Ga surface environment became oxidized remains debated (see summary in Steadman et al., 2020). Considering the last sentence is not the focus of this study, we have deleted it to avoid potential confusion from readers.

References:

- Holland HD. Volcanic gases, black smokers, and the Great Oxidation Event. *Geochimica et Cosmochimica acta* **66**, 3811–3826 (2002).
- Brounce M, Stolper E, Eiler J. Redox variations in Mauna Kea lavas, the oxygen fugacity of the Hawaiian plume, and the role of volcanic gases in Earth’s oxygenation. *Proceedings of the National Academy of Sciences* **114**, 8997–9002 (2017).
- Kelley KA, Cottrell E. Water and the oxidation state of subduction zone magmas. *Science* **325**, 605–607 (2009).
- Steadman JA, *et al.* Evidence for elevated and variable atmospheric oxygen in the Precambrian. *Precambrian Research*, 105722 (2020).
- Stolper DA, Keller CB. A record of deep-ocean dissolved O₂ from the oxidation state of iron in submarine basalts. *Nature* **553**, 323–327 (2018).

“I think the data are excellent, the interpretations may even be correct, but the arguments in support of these interpretations need to be developed more carefully in the main text. It may be that this is significantly easier for the authors in a longer format manuscript.

I hope that you find my comments here helpful. Best wishes for all, I hope everyone is well.”

Thank the reviewer for the critical and constructive comments. We have extended several points of the manuscript we agree are important to develop the arguments. Adding these points indeed help improve the manuscript. We also appreciate allowance from the editor to extend the manuscript to a longer format that satisfies the requirement of Nature Communications.

We sincerely wish you, the reviewers, and the editor the best and hope that you are all staying well and healthy during the challenging times!

REVIEWERS' COMMENTS

Reviewer #2 (Remarks to the Author):

Thank you for your detailed response, my qualms have been absolved. I do echo Dr. Brounce in saying that this doesn't seem to be the right journal for a paper with this much to say and a supplement this long is not a good compromise. But my scientific concerns have been dealt with and I think the paper should be published now.

-Willie Nicklas
Scripps Institution of Oceanography
UCSD

Reviewer #3 (Remarks to the Author):

I have read the revised manuscript, “Oxidized, sulfur-rich arc magmas formed porphyry Cu deposits by 1.88 Ga” by Xuyang Meng and co-authors.

1. Minor comment re: the Nicklas review

I don't think these authors have an issue with beam damage, so I think this comment from Nicklas is unwarranted. However, I point out to Meng et al. that beam *energy* is not the reason why. Even for mineral phases, even for S XANES analysis, you want to limit the *photon density* flux on the sample surface to be confident that you are avoiding beam damage. See Cottrell et al. 2018 for specific recommendations, but in my measurements of S-in-apatite, a photon flux on the order of 10^{10} /second with a ~5 um beam is more than safe.

2. A longer manuscript

I am glad that more text could be included in the main manuscript – I think this revision is significantly improved by the additions of a few sentences in various places throughout at the request of all three reviewers. I continue to seek quantitative treatment of some statements, but the authors have well-reasoned rebuttals to my points here. It is at their discretion.

Overall, I commend the authors on the revision. I look forward to seeing this in print at Nat Comm soon.

Sincerely,

Maryjo Brounce

Assistant Professor

RESPONSE TO REVIEWERS

Oxidized sulfur-rich arc magmas formed porphyry Cu deposit by 1.88 Ga

Response to Reviewer #2's comment:

"Thank you for your detailed response, my qualms have been absolved. I do echo Dr. Brounce in saying that this doesn't seem to be the right journal for a paper with this much to say and a supplement this long is not a good compromise. But my scientific concerns have been dealt with and I think the paper should be published now.

-Willie Nicklas

Scripps Institution of Oceanography, UCSD"

We appreciate your recommendation of publication. Thanks for your time in making the constructive comments. As verified by the editors, the lengths of the Methods and Supplementary Information satisfy the requirement of Nature Communications.

Response to Reviewer #3's comment:

We appreciate Dr. Maryjo Brounce for reviewing the revised version of our manuscript and the recommendation of publication in Nature Communications. See our replies as follows.

"I have read the revised manuscript, "Oxidized, sulfur-rich arc magmas formed porphyry Cu deposits by 1.88 Ga" by Xuyang Meng and co-authors.

1. Minor comment re: the Nicklas review

I don't think these authors have an issue with beam damage, so I think this comment from Nicklas is unwarranted. However, I point out to Meng et al. that beam energy is not the reason why. Even for mineral phases, even for S XANES analysis, you want to limit the *photon density flux* on the sample surface to be confident that you are avoiding beam damage. See Cottrell et al. 2018 for specific recommendations, but in my measurements of S-in-apatite, a photon flux on the order of 10¹⁰/second with a ~5 um beam is more than safe."

Thanks for the explanation of the recommended conditions to minimize beam damage. We agree that the photon density flux should be limited to avoid beam

damage and the beam energy is not the exact reason. We will be more meticulous in thinking and writing in the future, even for response letters.

“2. A longer manuscript

I am glad that more text could be included in the main manuscript – I think this revision is significantly improved by the additions of a few sentences in various places throughout at the request of all three reviewers. I continue to seek quantitative treatment of some statements, but the authors have well-reasoned rebuttals to my points here. It is at their discretion.

Overall, I commend the authors on the revision. I look forward to seeing this in print at Nat Comm soon.

Sincerely,

Maryjo Brounce

Assistant Professor”

We appreciate the constructive comments that have improved the first version of the manuscript. Thank you for recognition of our rebuttal to the suggestion of quantitative treatment of some statements.